# PROVABLY EFFICIENT FEDERATED ACTIVE MULTI-TASK REPRESENTATION LEARNING

## ABSTRACT

Multi-task representation learning is an emerging machine learning paradigm that integrates data from multiple sources, harnessing task similarities to enhance overall model performance. The application of multi-task learning to real-world settings is hindered due to data scarcity, along with challenges related to scalability and computational resources. To address these challenges, we develop a fast and sample-efficient approach for multi-task active learning with linear representation when the amount of data from source tasks and target tasks is limited. By leveraging the techniques from active learning, we propose an adaptive sampling-based alternating projected gradient descent (GD) and minimization algorithm that iteratively estimates the relevance of each source task to the target task and samples from each source task based on the estimated relevance. We present the convergence guarantees and the sample and time complexities of our algorithm. We evaluated the effectiveness of our algorithm using numerical experiments and compared it against four benchmark algorithms using synthetic and real MNIST-C datasets.

## 1 INTRODUCTION

Multi-task representation learning has emerged as a promising machine learning (ML) approach for simultaneously learning multiple related models by integrating data from various sources. The approach leverages shared structures between tasks to improve the performance of each individual task through collaboratively training similar yet different tasks to overcome a scarcity of data for any one task. This paradigm has been used with great success in natural language processing domains GPT-2 Radford et al. (2019), GPT-3 Brown et al. (2020), Bert Devlin et al. (2018), as well as vision domains CLIP (Radford et al., 2021). As noted in Radford et al. (2019), despite notable advances, existing learning systems require hundreds to thousands of examples to effectively induce functions that generalize well. With current approaches, this implies that multi-task training may need just as many effective training pairs to realize its potential. Most of the existing work on multi-task representation learning often assumes an unlimited number of samples for source tasks and a limited number of samples for the target task (Du et al., 2020; Chen et al., 2022). In practical applications such as medical imaging, drug discovery, fraud detection, and natural language processing in low-resource languages, data availability is limited, which restricts the application of existing ML approaches due to poor sample efficiency. It may be challenging to continue scaling the creation of datasets to the extent that might be necessary using current techniques. This motivates exploring new approaches for multi-task learning, specifically to develop provable methods that are fast and sample-efficient.

Additionally, as noted in Chen et al. (2022), not all tasks equally contribute to learning a representation. For instance, modern datasets like CIFAR-10, ImageNet, and the CLIP dataset were created using a list of search terms and a variety of different sources like search engines, news websites, and Wikipedia (Krizhevsky et al., 2009; Deng et al., 2009; Radford et al., 2021). Further, it is often unclear which tasks will best maximize performance on the target task.

In this paper, we aim to address these challenges by developing a federated active learning framework. Our goal is to learn a multi-task linear representation while prioritizing the relevance of the source tasks to generalize to a specific target task. Our approach involves using alternating gradient descent (GD) and minimization estimator to estimate the unknown parameters. Additionally, we utilize adaptive sampling to incorporate data samples from more relevant tasks into the learning process,

which benefits the generalization of the target task and sample efficiency. Our approach is federated since the tasks only share their estimate with the central server rather than the raw data itself.

**Our Contributions.** We introduce a novel active (adaptive) multi-task learning framework and an associated algorithm with guarantees. Next, we present our main contributions.

**(i) Active Low-Rank Representation Learning (A-LRRL) algorithm.** We adapt the alternating gradient descent (GD) and minimization approach of Lin et al. (2024); Nayer & Vaswani (2023); Collins et al. (2021) to provide a provable solution to the *active* multi-task representation learning problem. Our proposed A-LRRL algorithm presents an alternating GD and minimization estimator, which is fast, federated, and sample-efficient for learning the common low-dimensional representation. Our algorithm iteratively learns the unknown feature matrix. Using the learned representation, we further estimate the unknown relevance parameter and develop an adaptive sampling approach that samples the source task data based on the relevance estimate. Both the time and sample complexity of our solution depend only logarithmically on $1/\epsilon$.

**(ii) Convergence guarantees.** We present convergence guarantees for the proposed approach along with sample and time complexities. Our results show that the number of target samples only scales with the dimension/rank of the low-dimensional feature space and not on the input dimension to achieve $\epsilon-$accuracy in the excess risk for generalizing to the target task. Additionally, the number of source task samples in each epoch scales with $\max(\log d, \log M, k)\log(1/\epsilon)$. Here $d, M, k$ denote the input dimension, number of tasks, and low-dimensional feature dimension. Our main contribution is the convergence guarantee of the excess risk for two settings: (i) when the relevance parameter $\nu^\star$ is unknown and (ii) when the relevance parameter $\nu^\star$ is known. For both cases, we show that the sample complexity for the source tasks scales according to the sparsity of the relevance parameter. Hence the sample complexity of the proposed approach improves by a factor of the number of tasks compared to the naive uniform sampling approach. Our result is in agreement with that in (Chen et al., 2022). Further, we provide guarantees for the non-convex estimator. For the unknown setting, our result shows that the convergence guarantee is at least as good as that of the uniform sampling approach, and additionally, the sample complexity is as good as that of the known setting.

**(iii) Numerical performance.** We compared our framework with four benchmark approaches via simulations. We performed three experiments by varying the number of tasks, problem dimension, and rank of the feature matrix. We performed experiments on synthetic and real-world MNIST-C datasets in Mu & Gilmer (2019). The proposed approach consistently outperformed the benchmark algorithms in all cases. Thus, our experimental results validate the effectiveness of our approach.

## 2 NOTATIONS AND PROBLEM FORMULATION

**Notations.** We denote the set containing the first $n$ positive integers as $[n]$, which is defined as $\{1, 2, \ldots, n\}$. The $\ell_2$ norm of a vector $x$ is represented by $\|x\|$, while the spectral norm and the Frobenius norm of a matrix $A$ are denoted by $\|A\|$ and $\|A\|_F$, respectively. The transpose operation for matrices and vectors is indicated by $\top$, and $|x|$ refers to the element-wise absolute value of the vector $x$. The identity matrix of size $n \times n$ is denoted as $I_n$, often abbreviated as $I$, and $e_k$ denotes the $k$-th canonical basis vector, i.e., the $k$-th column of $I_n$. We define the $n_m$ i.i.d. samples from the $m$-th source task as an input matrix $X_m \in \mathbb{R}^{n_m \times d}$, with the corresponding output vector $Y_m \in \mathbb{R}^{n_m}$ and a noise vector $Z_m \in \mathbb{R}^{n_m}$. The vectors $\{w_m\}_{m \in [M]}$, is assembled into the matrix $W \in \mathbb{R}^{k \times M}$.

**Problem Formulation.** Consider $M$ source tasks and one target task, referred to as the $(M+1)$-th task. Every task $m \in [M+1]$ is associated with a distinct joint distribution $\mu_m$ over $\mathcal{X} \times \mathcal{Y}$, where $\mathcal{X} \in \mathbb{R}^d$ represents the input space and $\mathcal{Y} \in \mathbb{R}$ represents the output space. For each source task $m \in [M]$, we are given $n_m$ data samples $(x_{m,1}, y_{m,1}), \cdots, (x_{m,n_m}, y_{m,n_m})$, which are i.i.d. and sampled from the distribution $\mu_m$. The goal of multitask learning is to simultaneously produce predictive models for all $M$ source tasks, with the aim of finding common property among these tasks. We consider the existence of an underlying representation function $\phi^\star := \mathcal{X} \to \mathcal{Z}$, which transforms inputs into a feature space $\mathcal{Z} \in \mathbb{R}^k$ with $k \ll d$, within a specified set of functions $\Phi$ such as linear functions. Furthermore, we consider a linear transformation from the feature space to the output space, represented by the vector $w_m^\star \in \mathbb{R}^k$. Specifically, we assume that a sample $(x, y)$ from $\mu_m$ for any task $m \in [M+1]$ can be represented as $y = \phi^\star(x)^\top w_m^\star + z_m$, where $z_m$ is a noise.

*We consider a data-scarce regime, both for the source and the target task, i.e., $n_m < d$.* We consider a fixed amount of data for both source and target tasks, denoted as $\{(x_{m,1}, y_{m,1}), \cdots, (x_{m,n_m}, y_{m,n_m})\}_{m \in [M+1]}$ which is drawn i.i.d. from the task distributions $\mu_m$ for $m \in [M+1]$. We note that, typically, the number of data samples for the target task is even fewer than that of the source task, i.e., $n_{M+1} \ll \{n_1, \ldots, n_M\}$. This setting aligns with our main objective of active representation learning under scarce data, in which we have a limited amount of data available for the source task but have even less access to the target task data.

Define $\mathcal{L}_{M+1}(\phi, w) := \mathbb{E}_{(x,y)\sim\mu_{M+1}}[(\langle\phi(x), w\rangle - y)^2]$. The main objective is to use as few total samples from the source task as possible to learn a representation and linear predictor $\phi, w_{M+1}$ that effectively minimizes the excess risk on the target task, defined as

$$\mathrm{ER}_{M+1}(\phi, w) = \mathcal{L}_{M+1}(\phi, w) - \mathcal{L}_{M+1}(\phi^\star, w_{M+1}^\star). \tag{1}$$

We focus on the linear representation function class, studied in (Chen et al., 2022; Du et al., 2020; Tripuraneni et al., 2021; Cella & Pontil, 2021). We make the following low-dimensional assumption.

**Assumption 2.1** (Low-dimension linear representation). $\Phi = \{x \to B^\top x | B \in \mathbb{R}^{d \times k}\}$. We denote the true underlying representation function as $B^\star$.

The low-dimensional assumption captures the relatedness between the tasks and is used in many works on representation learning, including (Chen et al., 2022; Du et al., 2020; Tripuraneni et al., 2021; Yang et al., 2020; Hu et al., 2021; Cella et al., 2023; Kumar et al., 2022). Under Assumption 2.1, we can rewrite $y$ as $y = x^\top B^\star w_m^\star + z_m = x^\top \theta_m^\star + z_m$. We assume $\Theta^\star$ is a rank-$k$ matrix, where $k \ll \min\{d, M\}$. Let $\Theta^\star := [\theta_1^\star, \ldots, \theta_M^\star] \stackrel{SVD}{=} B^\star \Sigma^\star V^\star = B^\star W^\star$, denote the reduced (rank $k$) SVD, i.e., $B^\star$ and $V^{\star\top}$ are matrices with orthonormal columns, $B^\star$ is $d \times k$, $V^\star$ is $k \times M$, and $\Sigma$ is an $k \times k$ diagonal matrix with singular values. We let $W^\star := \Sigma V^\star$. We use $\sigma_{\max}^\star$ and $\sigma_{\min}^\star$ to denote the maximum and minimum singular values of $\Sigma$, and condition number $\kappa := \sigma_{\max}^\star / \sigma_{\min}^\star$.

Inspired by Chen et al. (2022), in our model, task relevance is a crucial factor. That is, we consider a setting where the goal is to learn a *specific* target task, rather than a *generic* target task as in (Du et al., 2020; Tripuraneni et al., 2021). Since $\sigma_{\min}(W^\star) > 0$, the coefficient $w_{M+1}^\star$ can be considered a linear combination of the coefficients $\{w_m^\star\}_{m \in [M]}$. Therefore, we make the assumption that $\nu^\star \in \mathbb{R}^M$, such that $W^\star \nu^\star = w_{M+1}^\star$, where a larger value of $|\nu^\star(m)|$ (the $m$-th element of the vector $\nu^\star$) indicates a stronger relevance for source task $m$ for the target task. Based on the information provided by $\nu^\star$, we prioritize samples from source tasks with the highest relevance. In this paper, we aim to learn the low-dimensional representation and the relevance parameter $\nu^\star$ to expedite collaborative learning among the source tasks and facilitate generalization to a target task.

**Assumption 2.2.** (Gaussian design & noise) We assume $x_{m,n}$ follows an i.i.d. standard Gaussian distribution and noise variables $z_m$ follow i.i.d. Gaussian distribution with zero mean and $\sigma^2$ variance.

We work in the random design linear regression setting, and in this context, Assumption 2.2 is standard (Chen et al., 2022; Du et al., 2020; Cella & Pontil, 2021; Tripuraneni et al., 2021).

**Assumption 2.3** (Incoherence). We assume that $\|w_m^\star\|^2 \leqslant \mu^2 \frac{k}{M} {\sigma_{\max}^\star}^2$ for a constant $\mu \geqslant 1$.

Recovering the feature matrix is impossible without any structural assumption. Notice that $y_m$s are not global functions of $\Theta^\star$, i.e., no $y_{m,n}$ is a function of the entire matrix $\Theta^\star$. We thus need an assumption that enables correct interpolation across the different columns. The incoherence (w.r.t. the canonical basis) assumption on the right singular vectors suffices for this purpose. Such an assumption on both left and right singular vectors was first introduced in Candes & Recht (2012) and used recently in representation learning (Tripuraneni et al., 2021; Collins et al., 2021; Thekumparampil et al., 2021).

## 3 RELATED WORK

**Multi-task representation learning** has been extensively explored, with roots traced back to seminal works such as (Caruana, 1997; Thrun & Pratt, 1998; Baxter, 2000). Many recent works studied provable non-adaptive multi-task representation learning under various assumptions. Du et al. (2020); Tripuraneni et al. (2021); Thekumparampil et al. (2021); Collins et al. (2021); Xu & Tewari (2021) focus on learning a representation function for *any* potential target task under the assumption of the

existence of a shared low-dimensional linear representation across all tasks. Recently, Wang et al. (2023); Chen et al. (2022) developed an adaptive representation learning for a specific target task under a similar setting as in (Du et al., 2020). Wang et al. (2023) improved the sample complexity on Chen et al. (2022) under a high dimension input assumption. There also exists works on empirical multi-task representation learning/transfer learning (Yao et al., 2022; Zamir et al., 2018).

While representation learning has achieved tremendous success, there remain challenges in providing theoretical guarantees. Most of the existing theoretical studies adopt a convex relaxation of the original non-convex problem and rely on the assumption that an optimal solution to the non-convex problem is known for their theoretical analysis (Du et al., 2020; Tripuraneni et al., 2021; Knight & Duan, 2024). The primary focus of these works is to demonstrate the dimensionality-reducing benefits of representation learning by showing that the number of target samples exceeds only $O(k)$, where $k$ is assumed to be small. Our work complements these results by showing how to provably and efficiently learn the representation in the linear case. The most closely related work to ours is (Chen et al., 2022). Our work extends and complements Chen *et al.* to tackle two key challenges: (1) The estimation approach in Chen et al. (2022) utilizes Du et al. (2020), which assumes an optimal solution to the non-convex estimation problem is available. In this work, we present a novel adaptive sampling-based alternating gradient descent and minimization-based estimator to solve the non-convex representation problem with generalization guarantees to a target task. (2) Chen et al. (2022); Du et al. (2020) considered that the number of source task samples must exceed the problem dimension $d$. We relax this assumption in our approach, and our guarantees hold for setting where the number of data samples is fewer than the problem size. Hence, our approach is viable for many practical applications with large problem sizes, however, with fewer data samples. This is specifically true in many image-related learning problems as validated through our simulations. In this work, we also consider the findings of Lin & Moothedath (2024), where the relevance parameter $\nu^\star$ is known, simplifying the problem to learning $\Theta^\star$. In the unknown setting, which is the main focus of this paper, the error in estimating $\nu^\star$ affects the estimation of $\Theta^\star$ in the next epoch, leading to a temporal error propagation. This requires new techniques to derive guarantees and ensure convergence.

**Matrix learning** is another related line of work in the low-rank matrix learning literature (Collins et al., 2021; Nayer & Vaswani, 2023). Collins et al. (2021); Nayer & Vaswani (2023) proposed an alternating GD minimization algorithm for recovering a low-rank matrix from compressed signals. The focus of these works is to learn the low-dimensional linear representation. Collins et al. (2021) provided guarantees on linear convergence relative to the initialization error. However, it does not offer guarantees for the initialization error itself. Further, the matrix learning analysis in Collins et al. (2021); Nayer & Vaswani (2023); Vaswani (2024) considered a non-noisy setting where the observed signals are not affected by noise. Additionally, these works focus on learning a low-dimensional representation using a non-adaptive data sampling of the source samples and not on the generalization of the target task and quantifying the excess risk for the target task. Our work focuses on *active* representation learning and generalizability to target tasks. Through theoretical analysis and numerical simulations, we showcase how the adaptive sampling approach enhances generalizability over uniform sampling. Further, our algorithm utilizes a spectral initialization approach Candes & Recht (2012); Nayer & Vaswani (2023); Vaswani (2024) with a truncation that carefully initializes our non-convex problem which is very crucial in obtaining the convergence guarantees for the optimal solution. Our guarantees thus enhance the results in Collins et al. (2021) by providing a convergent solution with initialization guarantee and sample and time complexities.

**Multi-task learning for sequential decision-making** has been studied in the context of bandit learning and reinforcement learning (RL). Multi-task learning in RL domains is studied in many works, including (Taylor & Stone, 2009; Parisotto et al., 2015; D'Eramo et al., 2024; Arora et al., 2020). D'Eramo et al. (2024) demonstrated that representation learning has the potential to enhance the rate of the approximate value iteration algorithm. Arora et al. (2020) proved that representation learning can reduce the sample complexity of imitation learning. Multi-task bandit learning is studied in many works, including (Lin et al., 2024; Hu et al., 2021; Yang et al., 2020; Cella et al., 2023). Yang et al. (2020); Cella et al. (2023) considered a convex relaxation-based approach to estimate the unknown parameter matrix, while Hu et al. (2021) proposed an optimism in the face of uncertainty approach. These works focus on the regret analysis of the sequential decision-making problem.

## 4 PROPOSED ALGORITHM: ACTIVE LOW-RANK REPRESENTATION LEARNING (A-LRRL)) VIA ALTERNATING GD AND MINIMIZATION

Our objective is to acquire a low-dimensional linear representation and task relevance estimation from the training samples (source tasks) through an adaptive sampling approach, allowing the utilization of more data from source tasks that are more relevant to the target task rather than a uniform sampling approach. The rationale is that by incorporating more samples from pertinent tasks, we can accelerate the learning process. To this end, our algorithm starts by drawing $\propto (\nu^\star(m))^2$ i.i.d. samples from the corresponding offline data for each source task $m \in [M]$. We partition the learning horizon into $\Gamma$ epochs. Using the source task samples in each epoch $i \in [\Gamma]$, we minimize the cost function

$$f_i(\widehat{B}^{(i)}, \widehat{W}^{(i)}) = \sum_{m=1}^{M} \sum_{n=1}^{n_m^i} \|y_{m,n} - x_{m,n}^\top \widehat{B}^{(i)} \widehat{w}_m^{(i)}\|^2, \tag{2}$$

where $\widehat{B}^{(i)} \in \mathbb{R}^{d \times k}$ and $\widehat{W}^{(i)} \in \mathbb{R}^{k \times M}$. Subsequently, we use the estimated parameter $\widehat{B}^{(i)}$ along with the sample for the target task to further optimize the cost function

$$\widehat{w}_{M+1}^{(i)} = \arg\min_w \|X_{M+1}^i{}^\top \widehat{B}_T^{(i)} w - Y_{M+1}^i\|^2. \tag{3}$$

Equation 3 via a least-squares solution yields the estimated parameter $\widehat{w}_{M+1}^{(i)}$ for the target task. Finally using the estimates $\widehat{W}^{(i)}, \widehat{w}_{M+1}^{(i)}$, we now solve the constrained least-squares problem to get the minimum-norm (unique) solution

$$\hat{\nu}_{i+1} = \arg\min_\nu \|\nu\|_2^2 \text{ such that } \widehat{W}^{(i)} \nu = \widehat{w}_{M+1}^{(i)}. \tag{4}$$

Using the relevance estimate $\hat{\nu}_{i+1}$, in the next epoch, we sample the source task data such that we utilize more samples from tasks that are more relevant to the specific target task. This observation is motivated from Chen et al. (2022) that demonstrated the benefit of adaptive relevance-based sampling over uniform sampling. In our theoretical analysis, we show that the estimate of the relevance parameter obtained at the end of each epoch $|\hat{\nu}_i(m)|$ is $c\epsilon_i/M$-close to the true parameter $|\nu^\star(m)|$.

Now, we will elaborate on our approach for solving equation 2. Recall that $n_m < d$ and $k \ll \{n_m, d\}$. The cost function in equation 2 is non-convex due to the rank constraint. Hence, it requires careful initialization. Thus, in the first epoch, we perform a spectral initialization (Chen & Candes, 2015; Nayer & Vaswani, 2023). The initialization process starts by extracting the top $k$ singular vector from

$$\widehat{\Theta}_{0,full} = \left[ (\frac{1}{n_1} X_1^{(1)\top} Y_1^{(1)}), \cdots, (\frac{1}{n_M} X_M^{(1)\top} Y_M^{(1)}) \right] = \sum_{m=1}^{M} \frac{1}{n_m^1} \sum_{n=1}^{n_m^1} x_{m,n} y_{m,n} e_m^\top,$$

where $X_m^{(1)}$ is the feature matrix obtained by concatenating the feature vectors of task $m$. The expected value of the $m-$th task represents $B^\star w_m^\star$ with $\mathbb{E}[\widehat{\Theta}_{0,full}] = B^\star W^\star$. However, the large magnitude of the sum of independent sub-exponential random variables restricts the ability to determine a bound for the $\|\widehat{\Theta}_{0,full} - B^\star W^\star\|$ within the desired sample complexity. To tackle this, we use the truncation method introduced in Chen & Candes (2015), starting with the top $k$ singular vectors of

$$\widehat{\Theta}_0 = \sum_{m=1}^{M} \sum_{n=1}^{n_m^1} x_{m,n} y_{m,n} e_m^\top \mathbb{1}_{\{y_{m,n}^2 \leqslant \alpha\}},$$

where $\alpha = \frac{\tilde{C}}{\sum_{m=1}^M n_m^1} \sum_{m=1,n=1}^{M,n_m^1} y_{m,n}^2$, $\tilde{C} = 9\kappa^2\mu^2$, and $y_{m,trunc}(\alpha) := Y_m^{(1)} \circ \mathbb{1}_{\{|Y_m^{(1)}| \leqslant \sqrt{\alpha}\}}$. Using Singular Value Decomposition (SVD), we obtain the top $k$ singular vectors from $\widehat{\Theta}_0$ to obtain our initial estimate $\widehat{B}_0$. This method effectively filters out large values while maintaining the remaining values and serves as a reliable initial step in accurately estimating parameters.

After the initialization phase, we perform an alternating GD and minimization step to estimate $\widehat{B}$ and $\widehat{W}$ by minimizing the cost function in 2. Each iteration consists of two stages: independently optimizing $\widehat{w}_m$ for each task via a least square minimization step, followed by a GD step to update

---

**Algorithm 1**: Active Low-Rank Representation Learning (A-LRRL) Algorithm

---

1: **Input:** Representation function class $\Phi$, multiplier for $\alpha$ in init step, $\tilde{C}$, GD step size, $\eta$, Number of iterations, $T$, number of epochs $\Gamma$

2: Initialize $\hat{\nu}_1 = [\frac{1}{M}, \cdots, \frac{1}{M}]$ and $\epsilon_i = 2^{-i}$

3: **for** $i = 1, 2, \ldots, \Gamma$ **do**

4:      Set $n_m^i = \beta \hat{\nu}(m)^2 \epsilon_i^{-2}$, where $\beta = 2500 M^2$

5:      For each task $m$, draw $n_m^i$ i.i.d samples from the corresponding dataset $\{X_m^i, Y_m^i\}_{m=1}^M$

6:      **Sample-split:** Partition the measurements and measure matrices into $2T+1$ equal-sized disjoint sets: one for initialization and $2T$ sets each for the iterations in each epoch. Denote these by $\{X_{m,\tau}^i, Y_{m,\tau}^i\}_{m=1}^M$, $\tau = 00$ (only for epoch 1), $01, \cdots 2T$.

7:      **if** $i = 1$ **then**

8:          **Spectral initialization:**

9:          Use $Y_m^{(1)} \equiv Y_{m,00}^{(1)}$, $X_m^{(1)} \equiv X_{m,00}^{(1)}$, set $\alpha = \frac{\tilde{C}}{\sum_{m=1}^M n_m^1} \sum_{n=1}^{n_m^1} y_{m,n}^2$

10:          $y_{m,trunc}(\alpha) := Y_m^{(1)} \circ \mathbb{1}_{\{|Y_m^{(1)}| \leqslant \sqrt{\alpha}\}}$ and $\widehat{\Theta}_0 := \sum_{m=1}^M \frac{1}{n_m^1} X_m^{(1)\top} y_{m,trunc}(\alpha) e_m^\top$

11:          Set $\widehat{B}^{(0)} \leftarrow$ top-$k$-singular-vectors of $\widehat{\Theta}_0$

12:      **end if**

13:      **AltGDmin iterations:**

14:      Set $\widehat{B}_0 \leftarrow \widehat{B}^{(i\text{-}1)}$

15:      **for** $\ell = 1$ to $T$ **do**

16:          Let $\widehat{B} \leftarrow \widehat{B}_{\ell-1}$

17:          **Update** $\widehat{w}_{m,\ell}, \widehat{\theta}_{m,\ell}$**:** For $m \in [M]$, $\widehat{w}_{m,\ell} \leftarrow (X_{m,\tau}^{(i)} \widehat{B})^\dagger Y_{m,\tau}^{(i)}$ and $\widehat{\theta}_{m,\ell} \leftarrow \widehat{B} \widehat{w}_{m,\ell}$

18:          **Gradient w.r.t** $\widehat{B}$**:** With $Y_m^{(i)} \equiv Y_{m,T+\tau}^{(i)}$, $X_m^{(i)} \equiv X_{m,T+\tau}^{(i)}$, compute $\nabla_{\widehat{B}} f(\widehat{B}, \widehat{W}_\ell) = \sum_{m=1}^M \frac{1}{n_m^i} X_m^{(i)\top} (X_m^{(i)} \widehat{B} \widehat{w}_{m,\ell} - Y_m^{(i)}) \widehat{w}_{m,\ell}^\top$

19:          **GD step:** Set $\widehat{B}^+ \leftarrow \widehat{B} - \eta \nabla_{\widehat{B}} f(\widehat{B}, \widehat{W}_\ell)$

20:          **Projection step:** Compute $\widehat{B}^+ \overset{QR}{=} B^+ R^+$ and set $\widehat{B}_\ell \leftarrow B^+$

21:      **end for**

22:      Set $\widehat{B}^{(i)} \leftarrow \widehat{B}_T$ and set $\widehat{W}^{(i)} \leftarrow \widehat{W}_T$

23:      Observe $n_{M+1}^i$ samples $X_{M+1}^{(i)}$ and $Y_{M+1}^{(i)}$ for the target task

24:      Compute $\widehat{w}_{M+1}^{(i)} = \arg\min_w \|X_{M+1}^{(i)\top} \widehat{B}^{(i)} w - Y_{M+1}^{(i)}\|^2$

25:      Estimate the relevance parameter as $\hat{\nu}_{i+1} = \arg\min_\nu \|\nu\|_2^2$    s.t.    $\widehat{W}^{(i)} \nu = \widehat{w}_{M+1}^{(i)}$

26: **end for**

---

$\widehat{B}$, utilizing the QR decomposition to obtain the updated matrix $B^+$, represented as $\widehat{B}^+ \overset{QR}{=} B^+ R^+$. Then, the estimate of $B^\star$ for the $i^{\text{th}}$ epoch is set as the orthonormal $B^+$ obtained using the QR decomposition (step 20 in Algorithm 1). We now compute the estimated parameter $\widehat{w}_{M+1}$ by minimizing the cost function in equation 3 using the least squares estimator. Finally, we solve the minimum-norm least squares problem in equation 4 to estimate the relevance parameter. The estimate of the relevance parameter serves as the sampling parameter in the next epoch. We sample the source task data for the next epoch $\propto (\hat{\nu}_i(m))^2$, giving more weightage to the more relevant task.

## 5 THEORETICAL RESULTS AND GUARANTEES

This section presents guarantees for excess risk and sample complexities for both source and target tasks, including scenarios where $\nu^\star$ is known Lin & Moothedath (2024). Although $\nu^\star$ is often unknown, certain applications, like predicting weather parameters at different locations, allow experts to determine the relevance of $M$ tasks (e.g., temperature, humidity, precipitation) to a forecasting task $(M+1)$, such as air quality. This analysis facilitates a comparison between the two scenarios.

### 5.1 GUARANTEES FOR ALGORITHM 1 (UNKNOWN $\nu^\star$) AND ALGORITHM 2 (KNOWN $\nu^\star$)

Algorithm 1 deals with the unknown $\nu^\star$ setting and Theorem 5.1 presents the guarantee. Algorithm 2 (given in Appendix C) presents the $\nu^\star$ known setting and Theorem 5.2 presents its guarantee. The primary distinction is that in Algorithm 1, the estimate of $\nu^\star$ relies on the estimate of $\Theta^\star$ (i.e.,

$B^\star, W^\star$) and the estimate of $w^\star_{M+1}$, which introduces a temporal effect in error propagation. In both cases, we demonstrate that, given the suitable sample complexity conditions for both the source and target tasks, the excess risk is bounded by $\epsilon$ with high probability.

**Theorem 5.1.** *Consider Assumptions 2.2 and 2.3 hold. For any $\epsilon' > 0$, $\delta, \delta' \in [0,1]$, $C > 1$, let $\sigma^2 \leqslant \frac{c\|\theta^\star_m\|^2}{\mu k^3 \kappa^6 n^i_m}$, $\eta = \frac{0.4}{\sigma^{\star 2}_{max}}$, and $T = C\kappa^2 \log\frac{1}{\epsilon}$. If $n^i_m \geqslant C\max(\log d, \log M, k)\log\frac{1}{\epsilon}$, $\sum_{m=1}^M n^i_m \geqslant C\kappa^6 \mu^2 (d+M)k(\kappa^2 k^2 + \log\frac{1}{\epsilon})$, and $\sum_{i=1}^\Gamma \sum_{m=1}^M n^i_m = N = O\left(\frac{(1+\delta')}{(1-\delta')^2}ks^\star_\Gamma \epsilon(\|\nu^\star\|^2_2 + \frac{1}{M})\log\frac{1}{\delta}\right)$ then with high probability Algorithm 1 guarantees that*

$$\mathrm{ER}(\widehat{B}, \widehat{w}_{M+1}) \leqslant \epsilon$$

*if the target task sample complexity $n_{M+1}$ is at least*

$$O\left(\frac{\sigma^2(k+\log\frac{1}{\delta})\epsilon^{-1}}{1-\delta'}\right),$$

*where $s^\star_\Gamma = (1-\gamma)\|\nu^\star\|^\Gamma_{0,\gamma} + \gamma M$, $\|\nu^\star\|^\Gamma_{0,\gamma} := \left|\left\{m : |\nu^\star(m)| > \sqrt{\gamma \frac{\|\nu^\star\|^2_2}{\sum_{m=1}^M n^\Gamma_m}}\right\}\right|$ for $\gamma \in [0,1]$, $\epsilon = \max(\epsilon', \epsilon_{noise})$, $\epsilon_{noise} = C\kappa^2\sqrt{NSR}$, $NSR := \frac{\sigma^2}{\min_m \|\theta^\star_m\|^2}$.*

We present the proof of Theorem 5.1 in Appendix B.2.

**Proof sketch.** The guarantees for the excess risk and sample complexities for the active learning problem studied in this paper are based on the estimation guarantee of the proposed estimator. The alternating GD and minimization estimator is guaranteed to achieve $\epsilon-$guarantee for estimating the unknown rank-$k$ feature matrix $\Theta^\star$ with high probability, for any $\epsilon > 0$, if for each epoch $i$, the total source task samples $\sum_{m=1}^M n^i_m \geqslant C\kappa^6 \mu^2(d+M)k(\kappa^2 k^2 + \log(1/\epsilon))$ and the number of samples from each source task $n^i_m \geqslant C\max(\log d, \log M, k)\log(1/\epsilon)$. Utilizing the convergence guarantee for $\widehat{B}$ and $\widehat{W}$, we then provided a guarantee for estimating the relevance parameter. We note that solving for $\nu$ is a minimum-norm least squares problem. In Lemma B.2, we show that under the $(\widehat{B}, \widehat{W})$ guarantee the estimate of the relevance parameter $|\hat{\nu}_i(m)|$ is $c\epsilon_i/M$ close to the true value $|\nu^\star(m)|$. Utilizing this and some linear algebra results and adopting some of the proof techniques from Chen et al. (2022) for our proposed alternating GD and minimization algorithm, we provide the convergence guarantee for excess risk. We present the details in the Appendix.

**Theorem 5.2.** *Consider Assumptions 2.2 and 2.3 hold. For any $\epsilon' > 0$, $\delta, \delta' \in [0,1]$, $C > 1$, let $\sigma^2 \leqslant \frac{c\|\theta^\star_m\|^2}{\mu k^3 \kappa^6 n_m}$, $\eta = \frac{0.4}{\sigma^{\star 2}_{max}}$, and $T = C\kappa^2 \log\frac{1}{\epsilon}$. If $n_m \geqslant C\max(\log d, \log M, k)\log\frac{1}{\epsilon}$, then with high probability, Algorithm 2 guarantees that*

$$\mathrm{ER}(\widehat{B}_T, \widehat{w}_{M+1}) \leqslant \epsilon,$$

*whenever the total sampling budget from all sources $N$ is at least*

$$O\left(\min\left\{\frac{(1+\delta')}{(1-\delta')^2}k\|\nu^\star\|^2_2 s^\star \epsilon \log\frac{1}{\delta}, (d+M)k(k^2 + \log\frac{1}{\epsilon})\right\}\right)$$

*and the number of target samples $n_{M+1}$ is at least $O\left(\frac{\sigma^2(k+\log\frac{1}{\delta})}{(1-\delta')}\epsilon^{-1}\right)$, where $s^\star = (1-\gamma)\|\nu^\star\|_{0,\gamma} + \gamma M$, $\|\nu^\star\|_{0,\gamma} := \left|\left\{m : |\nu^\star_m| > \sqrt{\gamma \frac{\|\nu^\star\|^2_2}{N}}\right\}\right|$ for $\gamma \in [0,1]$, $\epsilon = \max(\epsilon', \epsilon_{noise})$, $\epsilon_{noise} = C\kappa^2\sqrt{NSR}$, $NSR := \frac{\sigma^2}{\min_m \|\theta^\star_m\|^2}$.*

We present the proof of Theorem 5.2 in Appendix C.3.

## 5.2 DISCUSSION AND COMPLEXITIES

**Discussion on Theorem 5.1 and Theorem 5.2.** The sample complexity of the source tasks depends only logarithmically on $1/\epsilon$. Compared to the known case, in the unknown $\nu^\star$ setting, Algorithm 1 requires only an additional low-order term. The probability of our guarantees increases as the

number of target samples $n_{M+1}$ increases, and the number of target samples scales only with $k \ll d$. Theorems 5.1 and 5.2 show that the number of source samples required depends on the task relevance denoted by $s^\star$. Since $\sqrt{\frac{\|\nu^\star\|_2^2}{\sum_{m=1}^M n_m^i}}$ is of the order of $\epsilon$, for $\gamma \approx 1/M$, we have $\mathrm{ER}(\widehat{B}_T, \widehat{w}_{M+1}) \leqslant \epsilon$ by using only those source tasks with relevance $|\nu^\star(m)| \gtrsim \epsilon$. Let us consider two boundary cases: (i) $\nu^\star$ is a 1-sparse vector, i.e., the target task only depends on one source task, and (ii) $\nu^\star$ is a scaled vector $\mathbf{1}$ where $\mathbf{1}$ is a vector of all ones, i.e., all source tasks are equally relevant (uniform sampling). For $\gamma = 0$, (i) gives $s^\star = 1$ and (ii) gives $s^\star = M$. Thus, uniform sampling requires $M$ times more source data samples than (i), validating the effectiveness of the adaptive sampling approach. The result in Chen et al. (2022) for the $\nu^\star$ known setting requires that the total sampling budget from all sources $N$ is at least $\widetilde{O}\left((kd + kM + \log(\frac{1}{\delta}))\sigma^2 s^\star \|\nu\|_2^2 \epsilon^{-2}\right)$ and the number of target samples $n_{M+1}$ is at least $\widetilde{O}\left(\sigma^2(k + \log(\frac{1}{\delta}))\epsilon^{-2}\right)$. For the unknown setting they need an additional $\widetilde{O}(Mk^2 d\sigma\epsilon^{-1}\sqrt{s^\star})$ source task samples. Further, the guarantees in Chen et al. (2022) are under the assumption that an optimal solution to the non-convex cost function is known. We present Theorem 5.2, which provides guarantees on the excess risk using the proposed alternating GD and minimization estimator.

**Time and Communication Complexity.** To analyze the time complexity of a given epoch $i$, we first calculate the computation time for the initialization step. To calculate $\Theta_0$, we need a time of order $\sum_{m=1}^M n_m^1 d$. The time complexity of the $k$-SVD step $dMk$ times the number of iterations required. We notice that to obtain an initial estimate of the span of $B^\star$ that is $\delta_0$-accurate, where $\delta_0 = \frac{c}{\kappa^2\sqrt{k}}$, it is sufficient to use an order $\log(\kappa k)$ number of iterations. Thus, since $n_m^1 \geqslant k$, the total complexity of the initialization phase is $O(d(\sum_{m=1}^M n_m^1 + Mk)\log(\kappa k)) = O(\sum_{m=1}^M n_m^1 d \log \kappa k)$. The time required for each gradient computation is $\sum_{m=1}^M n_m^i dk$. The QR decomposition process requires a time complexity of order $dk^2$. Additionally, the time required to update the columns of matrix $W$ using the least squares method is $O(\sum_{m=1}^M n_m^i dk)$. The number of iterations of these steps for each epoch can be expressed as $T = O(\kappa^2 \log \frac{1}{\epsilon})$. Upon finishing the alternating GD minimization iterations, in every epoch, we solve the least squared estimator to calculate $\widehat{w}_{M+1}^{(i)}$ and $\hat{\nu}_{i+1}$, with a complexity of $O(n_{M+1}dk + k^2 M)$. Thus, the overall time complexity is $O(\sum_{m=1}^M n_m^1 d \log(\kappa k) + \max(\sum_{m=1}^M n_m^i dk, dk^2, \sum_{m=1}^M n_m^i dk) \cdot \Gamma \cdot T + (n_{M+1}dk + k^2 M) \cdot \Gamma) = O(\kappa^2 \Gamma \sum_{m=1}^M n_m^i dk \log(\frac{1}{\epsilon})\log(\kappa) + (n_{M+1}dk + k^2 M) \cdot \Gamma)$. The communication complexity for each task in each iteration is of the order of $dk$. Hence, the total is $O(dk \cdot \kappa \log \frac{1}{\epsilon})$.

# 6 SIMULATIONS

In this section, we present the numerical experiments that validate the effectiveness of our proposed algorithm on both synthetic and real-world MNIST-C datasets. While the proposed algorithm and guarantees are designed for linear representations, we conducted experiments on the C-MNIST dataset to evaluate the effectiveness of our approach on non-linear models. We performed a comparative analysis of our algorithm with four benchmark approaches: (i) the Method-of-Moments (MoM) estimator presented in Yang et al. (2020); Tripuraneni et al. (2021), (ii) the approach in Chen *et al.* Chen et al. (2022), (iii) our proposed estimator via a uniform sampling approach, (iv) the approach in Collins *et al.* (Collins et al., 2021). We performed experiments on synthetic and MNIST-C datasets, varying the number of tasks $M$ and the rank $k$. Furthermore, in experiments on synthetic data, we also varied the dimension $d$ in addition to the number of tasks $M$ and the rank $k$. We noticed that the proposed algorithm consistently outperforms all four benchmark approaches, validating the benefit of our proposed approach. We present some additional experiments and discussion in Appendix D.

## 6.1 DATASETS

**Synthetic data:** In our experimental setup for the synthetic data, we defined the default setting parameters as $n_m^i = 50, d = 100, k = 2, M = 80$. Notice that $n_m^i < d$, which captures the data-scarce setting. In the experiments, we varied one of the parameters by keeping others fixed to the default setting. The entries of matrix $B^\star$ were randomly generated by orthonormalizing an i.i.d. standard Gaussian matrix. Similarly, the entries of matrix $W^\star$ for the source tasks were randomly generated according to an i.i.d. Gaussian distribution. The task relevance parameter $\nu^\star$ was generated by assigning 20% of tasks a weight of 2, 60% of tasks set to 6, and the remaining 20% tasks to 10. Using the generated $\nu^\star$ and $W^\star$, we construct $w_{M+1}^\star := W^\star \nu^\star$ for the target task. The matrices $X_m$

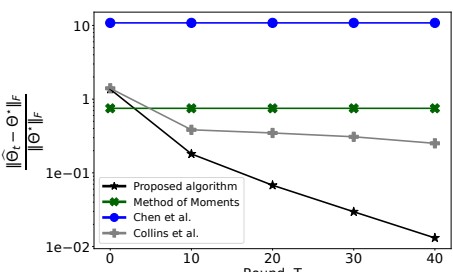

Figure 1: Estimation error vs. GD iteration for $d = 100$, $M = 80$, $k = 2$, noise variance $= 10^{-6}$.

were randomly generated using an i.i.d. standard Gaussian distribution. In addition, we utilized a noise model with a mean of zero and a variance of $10^{-6}$. It is important to note that in our experiments when we change the rank, number of source tasks, or dimensions, the matrices $B^\star$ and $W^\star$, as well as the data, are generated based on the specific dimensional setting of the problem. In this section, we present the simulation plots for the setting where the relevance parameter $\nu^\star$ is unknown. The plots for the known case of $\nu^\star$ are presented in Appendix D. The experimental results were derived by calculating the average of 100 independent Monte-Carlo trials.

**MNIST-C data:** In our experiment setup for the MNIST-C data, we evaluated our proposed algorithms on the corrupted MNIST dataset (MNIST-C) used in Mu & Gilmer (2019), which consists of 16 unique types of corruption. Although the MNIST problem is typically framed as a classification task with cross-entropy loss, we reformulate it as a regression problem with $\ell_2$ loss to align with the setting studied in this paper. To generate source and target tasks, each corrupted sub-dataset was partitioned into 10 tasks through the application of one-hot encoding to labels 0 through 9, resulting in 160 tasks, each identified as "corruption type + label." For each task, we converted the label into a binary format of $1/0$ based on the correspondence between the image and the label. Each task contained $28 \times 28$ dimensional 6000 images, which were normalized before processing. Experimental results are presented for two specific target tasks: brightness_0 and glass_blur_2. In our experiments, we defined the default parameter settings as $n_m^i = 100$, $d = 28^2 = 784$, $M = 50$, and $k = 40$. We varied the rank and the number of source tasks to evaluate the performance of our proposed approach.

## 6.2 RESULTS AND DISCUSSIONS

**Estimation error plot.** In Figure 1, we present the plot for estimation error vs. GD iterations for the first epoch. The MoM estimator is a noniterative method; hence, the estimation error is a single line. Chen et al. (2022) considered a convex relaxed solution of the original non-convex problem via the projected gradient descent method to obtain the estimation. Collins et al. (2021), on the other hand, does not provide an initialization guarantee, which affects the estimation again due to the non-convexity of the problem. It is difficult to obtain guarantees for the non-convex problem if the initialization error is not sufficiently small. The estimation error for the parameter matrix for the $M$ tasks $\Theta^\star$ is very low in our proposed estimator and it outperforms all the benchmark approaches. We note that the estimation error cannot be less than the noise variance, which is set to $10^{-6}$. This validates the benefit of adaptive sampling for generalizing to a target task.

**Excess risk plots.** Figure 2 presents the plots for the excess risk. Figures 2a, 2b, and 2c illustrate the excess risk for five algorithms as the number of source tasks $M$, rank $k$, and dimension $d$ vary for synthetic data. Similarly, Figures 2d, 2e, 2f, and 2g display the excess risk for the same algorithms while varying $M$ and $k$ for two target tasks from the MNIST-C dataset. We notice that our proposed approach outperforms the MoM estimator-based approach and the approach in Chen *et al.*. This is because, as also noted in Chen et al. (2022), during the iterative estimation process, the estimation error propagates from round to round due to unknown $\nu^\star$. Since the MoM estimator and the convex-relaxation approach in Chen et al. (2022) have considerable errors in the estimation of $\Theta^\star$, it negatively affects the estimation of $\nu^\star$. Our adaptive sampling approach slightly outperforms the uniform sampling method. We note that the benefit of adaptive sampling is majorly in the sample complexity while ensuring no worse convergence error guarantee compared to uniform sampling. Our approach also outperforms Collins *et al.* This is expected since the guarantees there depend on the initialization error; however, there is no initialization guarantee. Thus, the numerical experiments validate our theoretical findings and the effectiveness of our approach.

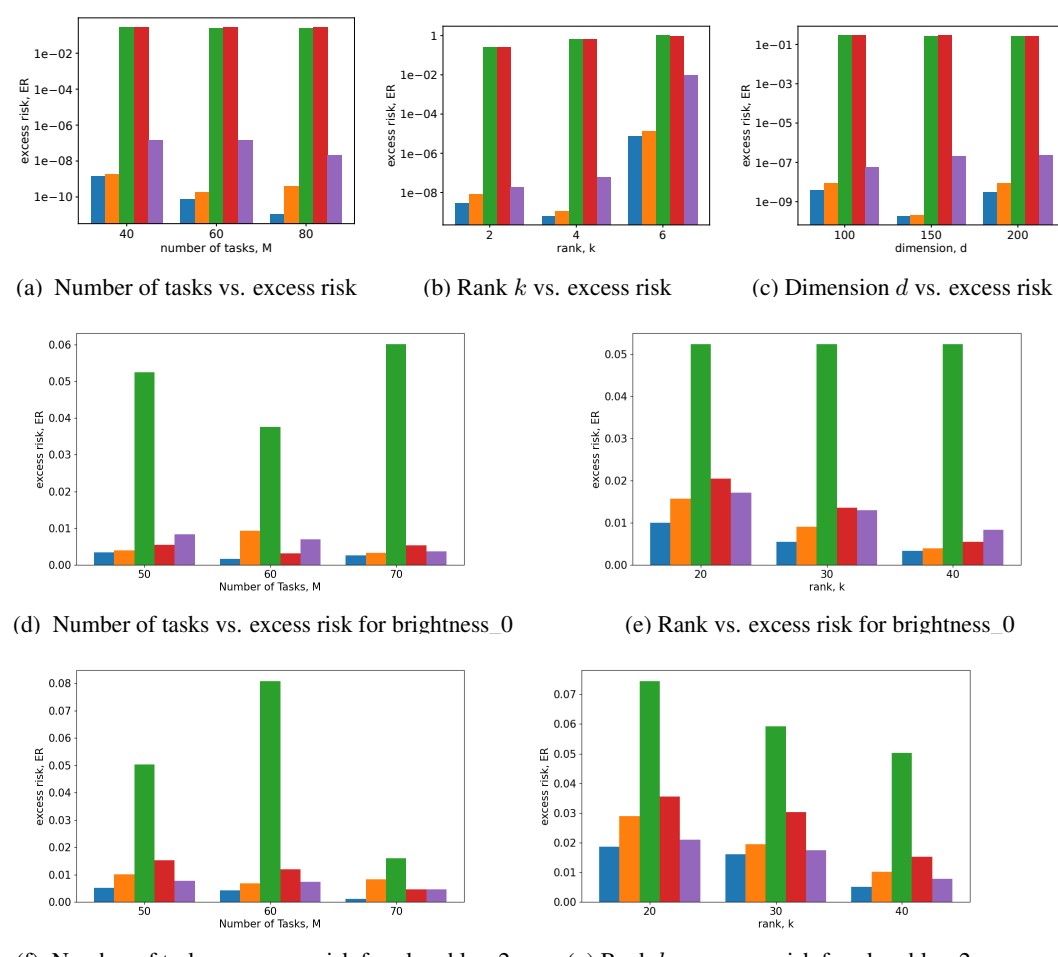

(a) Number of tasks vs. excess risk     (b) Rank $k$ vs. excess risk     (c) Dimension $d$ vs. excess risk

(d) Number of tasks vs. excess risk for brightness_0     (e) Rank vs. excess risk for brightness_0

(f) Number of tasks vs. excess risk for glass blur_2     (g) Rank $k$ vs. excess risk for glass blur_2

Figure 2: ■ proposed algorithm (adaptive sampling), ■ proposed algorithm (uniform sampling), ■ MoM (adaptive sampling), ■ Chen et al. (adaptive sampling), ■ Collins et al. (adaptive sampling). **Synthetic data:** We considered 50 data samples for each source task and 30 data samples for the target task. We varied the number of tasks as $M = 40, 60, 80$, varied the rank of the $\Theta^\star$ as $k = 2, 4, 6$, and varied the dimension as $d = 100, 150, 200$. As shown in the plots (Figures 2a, 2b, and 2c), our proposed approach with adaptive sampling outperforms the existing approaches. **MNIST-C data:** We considered 100 data samples for each source task and 50 data samples for the target task. We varied the number of tasks as $M = 50, 60, 70$, varied the rank of the $\Theta^\star$ as $k = 20, 30, 40$. The plot for MNIST-C data are presented in Figures 2d, 2e, 2f, and 2g.

## 7 CONCLUSION AND FUTURE WORK

In this work, we introduced a novel active-representation learning algorithm using an adaptive sampling-based alternating GD and minimization approach. Our proposed algorithm is specifically designed for active multi-task representation learning by considering the *unknown* task relevance to enable adaptive sampling. Our proposed approach can handle data-scarce settings where the number of source data samples is fewer than the problem dimension. We have demonstrated the algorithm's convergence guarantee in estimating the unknown feature matrix and the unknown relevance parameter. Additionally, we have evaluated the effectiveness of our approach in comparison with benchmark algorithms. The results clearly show that our proposed algorithm outperforms the benchmark approaches, thus validating its advantage over existing methods. As part of our future work, we aim to enhance our estimator by exploring alternative definitions of relevance parameters and adapting it to handle non-i.i.d. data scenarios effectively. Inspired by promising empirical results on nonlinear models, we plan to extend our approach to nonlinear representations in future work.

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

APPENDIX

## A  PRELIMINARIES

**Proposition A.1** (Theorem 2.8.1, (Vershynin, 2018))**.** *Let $X_1, \cdots, X_N$ be independent, mean zero, sub-exponential random variables. Then, for every $g \geqslant 0$, we have*

$$\mathbb{P}\Big\{ |\sum_{i=1}^N X_i \geqslant g| \Big\} \leqslant 2 \exp\left[ -c \min\left( \frac{g^2}{\sum_{i=1}^N \|X_i\|_{\psi_1}^2}, \frac{g}{\max_i \|X_i\|_{\psi_1}^2} \right) \right],$$

*where $c > 0$ is an absolute constant.*

**Proposition A.2** ((Kovanic, 1979))**.** *If $V$ is an $n \times n$ symmetrical matrix and if $X$ is an arbitrary $n \times q$ real matrix, then*

$$(V + XX^\top)^\dagger = V^\dagger - V^\dagger X(I + X^\top V^\dagger X)^{-1} X^\top V^\dagger + (X_\perp^\dagger)^\top X_\perp^\dagger,$$

*where $X_\perp = (I - VV^\dagger)X$.*

## B  THEORETICAL ANALYSIS OF ALGORITHM 1 ($\nu^\star$ UNKNOWN SETTING)

In this section, we present the details for the $\nu^\star$ unknown setting. Our goal is to provide guarantees for the excess risk and sample complexities for the source and target tasks, and time complexity. We first present the complete statement of Theorem 5.1.

**Complete form of Theorem 5.1:** Consider Assumptions 2.2 and 2.3 hold. For any $\epsilon' > 0, \delta, \delta' \in [0, 1], 0 < c < 1, C > 1$, let $\sigma^2 \leqslant \frac{c\|\theta_m^\star\|^2}{\mu k^3 \kappa^6 n_m^1}, \eta = \frac{0.4}{\sigma_{\max}^{\star 2}}$, and $T = C\kappa^2 \log \frac{1}{\epsilon}$. If

$$n_m^i \geqslant C \max(\log d, \log M, k) \log \frac{1}{\epsilon},$$

$$\sum_{m=1}^M n_m^i \geqslant C\kappa^6 \mu^2 (d + M)k(\kappa^2 k^2 + \log \frac{1}{\epsilon}), \text{ and}$$

$$\sum_{i=1}^\Gamma \sum_{m=1}^M n_m^i = N = O\left( \frac{(1 + \delta')}{(1 - \delta')^2} k s_\Gamma^\star \epsilon (\|\nu^\star\|_2^2 + \frac{1}{M}) \log \frac{1}{\delta} \right)$$

then with probability at least $1 - 2\delta - \Gamma T d^{-10} - 2de^{-\frac{\delta'^2 n_{M+1}}{3S}} - \exp(k - c\, n_{M+1})$, Algorithm 1 guarantees that

$$\text{ER}(\widehat{B}, \widehat{w}_{M+1}) \leqslant \epsilon$$

if the target task sample complexity $n_{M+1}$ is at least

$$n_{M+1} \geqslant \frac{\sigma^2 (2k + 3\log \frac{1}{\delta})\epsilon^{-1}}{2(1 - c)(1 - \delta')},$$

where $s_\Gamma^\star = (1 - \gamma)\|\nu^\star\|_{0,\gamma}^\Gamma + \gamma M, \|\nu^\star\|_{0,\gamma}^\Gamma := \left| \left\{ m : |\nu^\star(m)| > \sqrt{\gamma \frac{\|\nu^\star\|_2^2}{\sum_{m=1}^M n_m^\Gamma}} \right\} \right|$ for $\gamma \in [0, 1]$, $\epsilon = \max(\epsilon', \epsilon_{noise}), \epsilon_{noise} = C\kappa^2 \sqrt{NSR}, NSR := \frac{\sigma^2}{\min_m \|\theta_m^\star\|^2}$.

The notation $S$ in the theorem statement denotes an upper bound on the squared norm of the data vector, which is defined in Lemma C.1.

### B.1  SUPPORTING RESULTS AND PROOFS

**Lemma B.1.** *For any given value of $i$ and $m$, the following inequality holds:*

$$\left| (B^\star w_m^\star)^\top \left( ((B^\star W^\star)(B^\star W^\star)^\top)^\dagger - ((\widehat{B}^{(i)}\widehat{W}^{(i)})(\widehat{B}^{(i)}\widehat{W}^{(i)})^\top)^\dagger \right) B^\star w_{M+1}^\star \right|$$

$$\leqslant \|w_m^\star\|_2 \|w_{M+1}^\star\|_2 \|(\widehat{W}^{(i)}(\widehat{W}^{(i)})^\top)^\dagger\|_F \|\Delta_i\|_F \left( \|(\widehat{W}^{(i)}(\widehat{W}^{(i)})^\top)^\dagger\|_F \|\Delta_i\|_F + 2\|(\widehat{W}^{(i)})^\dagger\|_F \right).$$

*Proof.* Let us define $\Delta_i = B^\star W^\star - \widehat{B}^{(i)}\widehat{W}^{(i)}$. First, we perform an analysis of the term $((B^\star W^\star)(B^\star W^\star)^\top)^\dagger - ((\widehat{B}^{(i)}\widehat{W}^{(i)})(\widehat{B}^{(i)}\widehat{W}^{(i)})^\top)^\dagger$. We have

$$((B^\star W^\star)(B^\star W^\star)^\top)^\dagger - ((\widehat{B}^{(i)}\widehat{W}^{(i)})(\widehat{B}^{(i)}\widehat{W}^{(i)})^\top)^\dagger$$

$$= \left((\widehat{B}^{(i)}\widehat{W}^{(i)} + \Delta_i)(\widehat{B}^{(i)}\widehat{W}^{(i)} + \Delta_i)^\top\right)^\dagger - \left((\widehat{B}^{(i)}\widehat{W}^{(i)})(\widehat{B}^{(i)}\widehat{W}^{(i)})^\top\right)^\dagger$$

$$= \left((\widehat{B}^{(i)}\widehat{W}^{(i)})(\widehat{B}^{(i)}\widehat{W}^{(i)})^\top + \left(\Delta_i\Delta_i^\top + \Delta_i(\widehat{B}^{(i)}\widehat{W}^{(i)})^\top + (\widehat{B}^{(i)}\widehat{W}^{(i)})\Delta_i^\top\right)\right)^\dagger - \left((\widehat{B}^{(i)}\widehat{W}^{(i)})(\widehat{B}^{(i)}\widehat{W}^{(i)})^\top\right)^\dagger.$$

Assume that $V := (\widehat{B}^{(i)}\widehat{W}^{(i)})(\widehat{B}^{(i)}\widehat{W}^{(i)})^\top$, and proceed with decomposing $\left(\Delta_i\Delta_i^\top + \Delta_i(\widehat{B}^{(i)}\widehat{W}^{(i)})^\top + (\widehat{B}^{(i)}\widehat{W}^{(i)})\Delta_i^\top\right) =: XX^\top$. Let us assume that $\widehat{B}^{(i)}\widehat{W}^{(i)}$ is decomposed using singular value decomposition, represented as $U\Sigma\bar{V}^\top$. We have

$$VV^\dagger = U\Sigma^2 U^\top (U\Sigma^2 U^\top)^\dagger = UU^\top.$$

Therefore,

$$X_\perp^\dagger \widehat{B}^{(i)} = ((I - VV^\dagger)X)^\dagger \widehat{B}^{(i)} = ((I - UU^\top)X)^\dagger \widehat{B}^{(i)}$$

$$= (U_\perp U_\perp^\top X)^\dagger \widehat{B}^{(i)} \tag{5}$$

$$= X^\dagger (U_\perp U_\perp^\top)^\dagger \widehat{B}^{(i)} \tag{6}$$

$$= X^\dagger (U_\perp U_\perp^\top)^{-1} \widehat{B}^{(i)} \tag{7}$$

$$= X^\dagger U_\perp U_\perp^\top \widehat{B}^{(i)} \tag{8}$$

$$= 0 \tag{9}$$

where equation 5 can be derived from $UU^\top + U_\perp U_\perp^\top = I$ for an orthonormal matrix $U$. equation 6 can be derived from $(AB)^\dagger = B^\dagger A^\dagger$. equation 7 can be derived from $A^\dagger = A^{-1}$ for any positive definite matrix $A$. equation 8 can be derived from the fact that $U_\perp$ is an orthonormal matrix. equation 9 follows since $U, \widehat{B}^{(i)}\widehat{W}^{(i)}$ have the same column space and $\widehat{B}^{(i)}\widehat{W}^{(i)}, \widehat{B}^{(i)}$ have the same column space, we can conclude $U, \widehat{B}^{(i)}$ have the same column space. Thus $U_\perp^\top \widehat{B}^{(i)} = 0$. By applying Proposition A.2 and take into account the fact that $(I + X^\top V^\dagger X)^{-1} \preceq I$, we can derive

$$\left| (B^\star w_m^\star)^\top \left(((B^\star W^\star)(B^\star W^\star)^\top)^\dagger - ((\widehat{B}^{(i)}\widehat{W}^{(i)})(\widehat{B}^{(i)}\widehat{W}^{(i)})^\top)^\dagger\right) B^\star w_{M+1}^\star \right|$$

$$= \left| (B^\star w_m^\star)^\top \left(((B^\star W^\star)(B^\star W^\star)^\top)^\dagger - ((\widehat{B}^{(i)}\widehat{W}^{(i)})(\widehat{B}^{(i)}\widehat{W}^{(i)})^\top)^\dagger\right) \widehat{B}^{(i)}(\widehat{B}^{(i)})^\top B^\star w_{M+1}^\star \right|$$

$$= \left| (B^\star w_m^\star)^\top \left(((\widehat{B}^{(i)}\widehat{W}^{(i)})(\widehat{B}^{(i)}\widehat{W}^{(i)})^\top)^\dagger \left(\Delta_i\Delta_i^\top + \Delta_i(\widehat{B}^{(i)}\widehat{W}^{(i)})^\top + (\widehat{B}^{(i)}\widehat{W}^{(i)})\Delta_i^\top\right)((\widehat{B}^{(i)}\widehat{W}^{(i)})(\widehat{B}^{(i)}\widehat{W}^{(i)})^\top)^\dagger\right) B^\star w_{M+1}^\star \right|$$

$$\leqslant \|(B^\star w_m^\star)^\top\| \|((\widehat{B}^{(i)}\widehat{W}^{(i)})(\widehat{B}^{(i)}\widehat{W}^{(i)})^\top)^\dagger \left(\Delta_i\Delta_i^\top + \Delta_i(\widehat{B}^{(i)}\widehat{W}^{(i)})^\top + (\widehat{B}^{(i)}\widehat{W}^{(i)})\Delta_i^\top\right)((\widehat{B}^{(i)}\widehat{W}^{(i)})(\widehat{B}^{(i)}\widehat{W}^{(i)})^\top)^\dagger\| \|B^\star w_{M+1}^\star\|$$

$$\leqslant \|(B^\star w_m^\star)^\top\|_2 \|((\widehat{B}^{(i)}\widehat{W}^{(i)})(\widehat{B}^{(i)}\widehat{W}^{(i)})^\top)^\dagger \Delta_i\Delta_i^\top ((\widehat{B}^{(i)}\widehat{W}^{(i)})(\widehat{B}^{(i)}\widehat{W}^{(i)})^\top)^\dagger\|_F \|B^\star w_{M+1}^\star\|_2$$

$$+ \|(B^\star w_m^\star)^\top\|_2 \|((\widehat{B}^{(i)}\widehat{W}^{(i)})(\widehat{B}^{(i)}\widehat{W}^{(i)})^\top)^\dagger \Delta_i(\widehat{B}^{(i)}\widehat{W}^{(i)})^\top ((\widehat{B}^{(i)}\widehat{W}^{(i)})(\widehat{B}^{(i)}\widehat{W}^{(i)})^\top)^\dagger\|_F \|B^\star w_{M+1}^\star\|_2$$

$$+ \|(B^\star w_m^\star)^\top\|_2 \|((\widehat{B}^{(i)}\widehat{W}^{(i)})(\widehat{B}^{(i)}\widehat{W}^{(i)})^\top)^\dagger (\widehat{B}^{(i)}\widehat{W}^{(i)})\Delta_i^\top ((\widehat{B}^{(i)}\widehat{W}^{(i)})(\widehat{B}^{(i)}\widehat{W}^{(i)})^\top)^\dagger\|_F \|B^\star w_{M+1}^\star\|_2.$$

Given that

$$\|(\widehat{B}^{(i)}\widehat{W}^{(i)})^\top((\widehat{B}^{(i)}\widehat{W}^{(i)})(\widehat{B}^{(i)}\widehat{W}^{(i)})^\top)^\dagger\|_F = \|((\widehat{B}^{(i)}\widehat{W}^{(i)})(\widehat{B}^{(i)}\widehat{W}^{(i)})^\top)^\dagger (\widehat{B}^{(i)}\widehat{W}^{(i)})\|_F$$

$$= \|(\widehat{B}^{(i)}\widehat{W}^{(i)}(\widehat{W}^{(i)})^\top(\widehat{B}^{(i)})^\top)^\dagger(\widehat{B}^{(i)}\widehat{W}^{(i)})\|_F$$

$$= \|((\widehat{B}^{(i)})^\top)^\dagger(\widehat{W}^{(i)}(\widehat{W}^{(i)})^\top)^\dagger(\widehat{B}^{(i)})^\dagger(\widehat{B}^{(i)}\widehat{W}^{(i)})\|_F$$

$$= \|\widehat{B}^{(i)}(\widehat{W}^{(i)}(\widehat{W}^{(i)})^\top)^\dagger\widehat{W}^{(i)}\|_F \tag{10}$$

$$= \|(\widehat{W}^{(i)}(\widehat{W}^{(i)})^\top)^\dagger\widehat{W}^{(i)}\|_F \tag{11}$$

$$= \|((\widehat{W}^{(i)})^\top)^\dagger((\widehat{W}^{(i)})^\dagger\widehat{W}^{(i)})\|_F \tag{12}$$

$$= \|((\widehat{W}^{(i)})^\dagger\widehat{W}^{(i)})^\top(\widehat{W}^{(i)})^\dagger\|_F \tag{13}$$

$$= \|(\widehat{W}^{(i)})^\dagger\widehat{W}^{(i)}(\widehat{W}^{(i)})^\dagger\|_F \tag{14}$$

$$= \|(\widehat{W}^{(i)})^\dagger\|_F \tag{15}$$

where equation 10 and equation 11 can be derived from the fact that $\widehat{B}^{(i)}$ is a unitary matrix. equation 12 can be derived from $(AB)^{\dagger} = B^{\dagger}A^{\dagger}$. equation 13 can be derived from $\|A\|_F = \|A^{\top}\|_F$. equation 14 can be derived from $(A^{\dagger}A)^{\top} = A^{\dagger}A$. equation 15 can be derived from $A^{\dagger}AA^{\dagger} = A^{\dagger}$. Therefore, the final bound can be expressed as

$$\left| (B^{\star}w_m^{\star})^{\top}\left( ((B^{\star}W^{\star})(B^{\star}W^{\star})^{\top})^{\dagger} - ((\widehat{B}^{(i)}\widehat{W}^{(i)})(\widehat{B}^{(i)}\widehat{W}^{(i)})^{\top})^{\dagger} \right) B^{\star}w_{M+1}^{\star} \right|$$

$$\leqslant \|(B^{\star}w_m^{\star})^{\top}\|_2 \|((\widehat{B}^{(i)}\widehat{W}^{(i)})(\widehat{B}^{(i)}\widehat{W}^{(i)})^{\top})^{\dagger}\|_F^2 \|\Delta_i\|_F^2 \|B^{\star}w_{M+1}^{\star}\|_2$$

$$+ \|(B^{\star}w_m^{\star})^{\top}\|_2 \|((\widehat{B}^{(i)}\widehat{W}^{(i)})(\widehat{B}^{(i)}\widehat{W}^{(i)})^{\top})^{\dagger}\|_F \|\Delta_i\|_F \|(\widehat{W}^{(i)})^{\dagger}\|_F \|B^{\star}w_{M+1}^{\star}\|_2$$

$$+ \|(B^{\star}w_m^{\star})^{\top}\|_2 \|(\widehat{W}^{(i)})^{\dagger}\|_F \|\Delta_i\|_F \|((\widehat{B}^{(i)}\widehat{W}^{(i)})(\widehat{B}^{(i)}\widehat{W}^{(i)})^{\top})^{\dagger}\|_F \|B^{\star}w_{M+1}^{\star}\|_2$$

$$\leqslant \|w_m^{\star}\|_2 \|(\widehat{W}^{(i)}(\widehat{W}^{(i)})^{\top})^{\dagger}\|_F^2 \|\Delta_i\|_F^2 \|w_{M+1}^{\star}\|_2$$

$$+ 2\|w_m^{\star}\|_2 \|(\widehat{W}^{(i)}(\widehat{W}^{(i)})^{\top})^{\dagger}\|_F \|\Delta_i\|_F \|(\widehat{W}^{(i)})^{\dagger}\|_F \|w_{M+1}^{\star}\|_2$$

$$= \|w_m^{\star}\|_2 \|w_{M+1}^{\star}\|_2 \|(\widehat{W}^{(i)}(\widehat{W}^{(i)})^{\top})^{\dagger}\|_F \|\Delta_i\|_F \left( \|(\widehat{W}^{(i)}(\widehat{W}^{(i)})^{\top})^{\dagger}\|_F \|\Delta_i\|_F + 2\|(\widehat{W}^{(i)})^{\dagger}\|_F \right).$$

Thus, we complete the proof. $\qquad\square$

**Lemma B.2.** *Assume* $\sigma^2 \leqslant \frac{k}{M}\delta^{(i)2}\sigma_{\max}^{\star 2}$. *Set* $\eta = \frac{0.4}{\sigma_{\max}^{\star 2}}$, $\delta_0^2 \leqslant \frac{0.02}{C\mu^4\kappa^4 k^3}$ *and* $T \geqslant \frac{\log \frac{1}{\sqrt{2}}}{\log(1-\frac{0.4\mu c\eta}{\kappa^2})}$. *If*

$$\sum_{i=1}^{M} n_m^i \geqslant C\kappa^4\mu^2 dk \quad \text{and}$$

$$n_m^i \gtrsim \max(\log d, \log M, k),$$

*then with probability at least* $1 - \exp(d - \frac{cT\sum_{m=1}^{M}n_m^1}{\mu^6\kappa^8 k^5}) - 5iTM\exp(k - cn_m^i) - \exp(k - cn_{M+1})$, *it holds that*

$$|\nu^{\star}(m)| - \frac{0.02\epsilon_i}{M} \leqslant |\hat{\nu}_{i+1}(m)| \leqslant |\nu^{\star}(m)| + \frac{0.02\epsilon_i}{M}$$

*and*

$$|\hat{\nu}_{i+1}(m)| = \begin{cases} [\frac{1}{2}|\nu^{\star}(m)|, \frac{3}{2}|\nu^{\star}(m)|], & \text{if } |\nu^{\star}(m)| \geqslant \frac{0.02\epsilon_i}{M} \\ [0, \frac{0.06\epsilon_i}{M}], & \text{if } |\nu^{\star}(m)| \leqslant \frac{0.02\epsilon_i}{M} \end{cases}$$

*Proof.* Consider any epoch $i$ and its corresponding estimated representation $\widehat{B}^{(i)}$. Using the least squared method, we obtain

$$\widehat{w}_{M+1}^{(i)} = \arg\min_w \|X_{M+1}^{(i)}\widehat{B}^{(i)}w - Y_{M+1}\|^2$$

$$= ((X_{M+1}^{(i)}\widehat{B}^{(i)})^{\top}(X_{M+1}^{(i)}\widehat{B}^{(i)}))^{-1}(X_{M+1}^{(i)}\widehat{B}^{(i)})^{\top}Y_{M+1}^{(i)}$$

$$= ((X_{M+1}^{(i)}\widehat{B}^{(i)})^{\top}(X_{M+1}^{(i)}\widehat{B}^{(i)}))^{-1}(X_{M+1}^{(i)}\widehat{B}^{(i)})^{\top}(X_{M+1}^{(i)}B^{\star}w_{M+1}^{\star} + Z_{M+1}^{(i)})$$

$$= (\widehat{B}^{(i)\top}X_{M+1}^{(i)\top}X_{M+1}^{(i)}\widehat{B}^{(i)})^{-1}\widehat{B}^{(i)\top}X_{M+1}^{(i)\top}X_{M+1}^{(i)}B^{\star}w_{M+1}^{\star} + (\widehat{B}^{(i)\top}X_{M+1}^{(i)\top}X_{M+1}^{(i)}\widehat{B}^{(i)})^{-1}\widehat{B}^{(i)\top}X_{M+1}^{(i)\top}Z_{M+1}^{(i)}$$

By utilizing Lemma E.8 in Chen et al. (2022) and the aforementioned result, we derive

$$|\hat{\nu}_{i+1}(m)| = |\widehat{w}_m^{(i)\top}(\widehat{W}^{(i)}\widehat{W}^{(i)\top})^{\dagger}\widehat{w}_{M+1}^{(i)}|$$

$$= |(\widehat{B}^{(i)}\widehat{w}_m^{(i)})^{\top}((\widehat{B}^{(i)}\widehat{W}^{(i)})(\widehat{B}^{(i)}\widehat{W}^{(i)})^{\top})^{\dagger}(\widehat{B}^{(i)}\widehat{w}_{M+1}^{(i)})| \qquad (16)$$

$$= |(\widehat{B}^{(i)}\widehat{w}_m^{(i)})^{\top}((\widehat{B}^{(i)}\widehat{W}^{(i)})(\widehat{B}^{(i)}\widehat{W}^{(i)})^{\top})^{\dagger}\widehat{B}^{(i)}(\widehat{B}^{(i)\top}X_{M+1}^{(i)\top}X_{M+1}^{(i)}\widehat{B}^{(i)})^{-1}\widehat{B}^{(i)\top}X_{M+1}^{(i)\top}X_{M+1}^{(i)}B^{\star}w_{M+1}^{\star}$$

$$+ (\widehat{B}^{(i)}\widehat{w}_m^{(i)})^{\top}((\widehat{B}^{(i)}\widehat{W}^{(i)})(\widehat{B}^{(i)}\widehat{W}^{(i)})^{\top})^{\dagger}\widehat{B}^{(i)}(\widehat{B}^{(i)\top}X_{M+1}^{(i)\top}X_{M+1}^{(i)}\widehat{B}^{(i)})^{-1}\widehat{B}^{(i)\top}X_{M+1}^{(i)\top}Z_{M+1}^{(i)}|$$

$$= |(\widehat{B}^{(i)}\widehat{w}_m^{(i)})^{\top}((\widehat{B}^{(i)}\widehat{W}^{(i)})(\widehat{B}^{(i)}\widehat{W}^{(i)})^{\top})^{\dagger}\widehat{B}^{(i)}(\widehat{B}^{(i)\top}X_{M+1}^{(i)\top}X_{M+1}^{(i)}\widehat{B}^{(i)})^{-1}\widehat{B}^{(i)\top}X_{M+1}^{(i)\top}X_{M+1}^{(i)}\widehat{B}^{(i)}\widehat{B}^{(i)\top}B^{\star}w_{M+1}^{\star}$$

$$+ \widehat{w}_m^{(i)\top}(\widehat{W}^{(i)}\widehat{W}^{(i)\top})^{\dagger}M^{-1}\widehat{B}^{(i)\top}X_{M+1}^{(i)\top}Z_{M+1}^{(i)}| \qquad (17)$$

$$= |(B^{\star}w_m^{\star})^{\top}((\widehat{B}^{(i)}\widehat{W}^{(i)})(\widehat{B}^{(i)}\widehat{W}^{(i)})^{\top})^{\dagger}B^{\star}w_{M+1}^{\star} + (\widehat{B}^{(i)}\widehat{w}_m^{(i)} - B^{\star}w_m^{\star})^{\top}((\widehat{B}^{(i)}\widehat{W}^{(i)})(\widehat{B}^{(i)}\widehat{W}^{(i)})^{\top})^{\dagger}B^{\star}w_{M+1}^{\star}$$

$$+ \widehat{w}_m^{(i)\top}(\widehat{W}^{(i)}\widehat{W}^{(i)\top})^{\dagger}M^{-1}\widehat{B}^{(i)\top}X_{M+1}^{(i)\top}Z_{M+1}^{(i)}|$$

where equation 16 can be derived from the fact that $\widehat{B}^{(i)}$ is an orthonormal matrix. equation 17 can be derived from the fact that $\widehat{B}^{(i)}$ is an orthonormal matrix and $M = \widehat{B}^{(i)^\top} X_{M+1}^{(i)^\top} X_{M+1}^{(i)} \widehat{B}^{(i)}$. Thus, using the triangular inequality, we find

$$
\begin{aligned}
|\hat{\nu}_{i+1}(m)| \leqslant & |(B^\star w_m^\star)^\top ((\widehat{B}^{(i)}\widehat{W}^{(i)})(\widehat{B}^{(i)}\widehat{W}^{(i)})^\top)^\dagger B^\star w_{M+1}^\star| \\
& + |(\widehat{B}^{(i)}\widehat{w}_m^{(i)} - B^\star w_m^\star)^\top ((\widehat{B}^{(i)}\widehat{W}^{(i)})(\widehat{B}^{(i)}\widehat{W}^{(i)})^\top)^\dagger B^\star w_{M+1}^\star| \\
& + |\widehat{w}_m^{(i)^\top} (\widehat{W}^{(i)}\widehat{W}^{(i)^\top})^\dagger M^{-1} \widehat{B}^{(i)^\top} X_{M+1}^{(i)^\top} Z_{M+1}^{(i)}| \\
|\hat{\nu}_{i+1}(m)| \geqslant & |(B^\star w_m^\star)^\top ((\widehat{B}^{(i)}\widehat{W}^{(i)})(\widehat{B}^{(i)}\widehat{W}^{(i)})^\top)^\dagger B^\star w_{M+1}^\star| \\
& - |(\widehat{B}^{(i)}\widehat{w}_m^{(i)} - B^\star w_m^\star)^\top ((\widehat{B}^{(i)}\widehat{W}^{(i)})(\widehat{B}^{(i)}\widehat{W}^{(i)})^\top)^\dagger B^\star w_{M+1}^\star| \\
& - |\widehat{w}_m^{(i)^\top} (\widehat{W}^{(i)}\widehat{W}^{(i)^\top})^\dagger M^{-1} \widehat{B}^{(i)^\top} X_{M+1}^{(i)^\top} Z_{M+1}^{(i)}|
\end{aligned}
$$

Furthermore, according to Lemma E.8 in Chen et al. (2022), $\nu^\star(m) = w_m^{\star^\top} (W^\star W^{\star^\top})^{-1} w_{M+1}^\star = (B^\star w_m^\star)^\top ((B^\star W^\star)(B^\star W^\star)^\top)^\dagger (B^\star w_{M+1}^\star)$ holds. By applying the Cauchy–Schwarz inequality, Lemma B.1, and considering the combination of these results, we can prove

$$
\begin{aligned}
|\hat{\nu}_{i+1}(m)| - |\nu^\star(m)| \leqslant & |(B^\star w_m^\star)^\top ((\widehat{B}^{(i)}\widehat{W}^{(i)})(\widehat{B}^{(i)}\widehat{W}^{(i)})^\top)^\dagger B^\star w_{M+1}^\star| \\
& - |(B^\star w_m^\star)^\top ((B^\star W^\star)(B^\star W^\star)^\top)^\dagger (B^\star w_{M+1}^\star)| \\
& + |(\widehat{B}^{(i)}\widehat{w}_m^{(i)} - B^\star w_m^\star)^\top ((\widehat{B}^{(i)}\widehat{W}^{(i)})(\widehat{B}^{(i)}\widehat{W}^{(i)})^\top)^\dagger B^\star w_{M+1}^\star| \\
& + |\widehat{w}_m^{(i)^\top} (\widehat{W}^{(i)}\widehat{W}^{(i)^\top})^\dagger M^{-1} \widehat{B}^{(i)^\top} X_{M+1}^{(i)^\top} Z_{M+1}^{(i)}| \\
\leqslant & \left| (B^\star w_m^\star)^\top \left( ((B^\star W^\star)(B^\star W^\star)^\top)^\dagger - ((\widehat{B}^{(i)}\widehat{W}^{(i)})(\widehat{B}^{(i)}\widehat{W}^{(i)})^\top)^\dagger \right) B^\star w_{M+1}^\star \right| \\
& + |(\widehat{B}^{(i)}\widehat{w}_m^{(i)} - B^\star w_m^\star)^\top ((\widehat{B}^{(i)}\widehat{W}^{(i)})(\widehat{B}^{(i)}\widehat{W}^{(i)})^\top)^\dagger B^\star w_{M+1}^\star| \\
& + |\widehat{w}_m^{(i)^\top} (\widehat{W}^{(i)}\widehat{W}^{(i)^\top})^\dagger M^{-1} \widehat{B}^{(i)^\top} X_{M+1}^{(i)^\top} Z_{M+1}^{(i)}| \\
\leqslant & \|w_m^\star\|_2 \|w_{M+1}^\star\|_2 \|(\widehat{W}^{(i)}(\widehat{W}^{(i)})^\top)^\dagger\|_F \|\Delta_i\|_F \left( \|(\widehat{W}^{(i)}(\widehat{W}^{(i)})^\top)^\dagger\|_F \|\Delta_i\|_F + 2\|(\widehat{W}^{(i)})^\dagger\|_F \right) \\
& + \|w_{M+1}^\star\|_2 \|(\widehat{W}^{(i)}(\widehat{W}^{(i)})^\top)^\dagger\|_F \|\widehat{\theta}_m^{(i)} - \theta_m^\star\|_2 \\
& + \|(\widehat{W}^{(i)}(\widehat{W}^{(i)})^\top)^\dagger\|_F \|M^{-1}\| \|\widehat{w}_m^{(i)}\|_2 \|\widehat{B}^{(i)^\top} X_{M+1}^{(i)^\top} Z_{M+1}^{(i)}\|_2 \\
|\nu^\star(m)| - |\hat{\nu}_{i+1}(m)| \leqslant & |(B^\star w_m^\star)^\top ((B^\star W^\star)(B^\star W^\star)^\top)^\dagger (B^\star w_{M+1}^\star)| \\
& - |(B^\star w_m^\star)^\top ((\widehat{B}^{(i)}\widehat{W}^{(i)})(\widehat{B}^{(i)}\widehat{W}^{(i)})^\top)^\dagger B^\star w_{M+1}^\star| \\
& + |(\widehat{B}^{(i)}\widehat{w}_m^{(i)} - B^\star w_m^\star)^\top ((\widehat{B}^{(i)}\widehat{W}^{(i)})(\widehat{B}^{(i)}\widehat{W}^{(i)})^\top)^\dagger B^\star w_{M+1}^\star| \\
& + |\widehat{w}_m^{(i)^\top} (\widehat{W}^{(i)}\widehat{W}^{(i)^\top})^\dagger M^{-1} \widehat{B}^{(i)^\top} X_{M+1}^{(i)^\top} Z_{M+1}^{(i)}| \\
\leqslant & \left| (B^\star w_m^\star)^\top \left( ((B^\star W^\star)(B^\star W^\star)^\top)^\dagger - ((\widehat{B}^{(i)}\widehat{W}^{(i)})(\widehat{B}^{(i)}\widehat{W}^{(i)})^\top)^\dagger \right) (B^\star w_{M+1}^\star) \right| \\
& + |(\widehat{B}^{(i)}\widehat{w}_m^{(i)} - B^\star w_m^\star)^\top ((\widehat{B}^{(i)}\widehat{W}^{(i)})(\widehat{B}^{(i)}\widehat{W}^{(i)})^\top)^\dagger B^\star w_{M+1}^\star| \\
& + |\widehat{w}_m^{(i)^\top} (\widehat{W}^{(i)}\widehat{W}^{(i)^\top})^\dagger M^{-1} \widehat{B}^{(i)^\top} X_{M+1}^{(i)^\top} Z_{M+1}^{(i)}| \\
\leqslant & \|w_m^\star\|_2 \|w_{M+1}^\star\|_2 \|(\widehat{W}^{(i)}(\widehat{W}^{(i)})^\top)^\dagger\|_F \|\Delta_i\|_F \left( \|(\widehat{W}^{(i)}(\widehat{W}^{(i)})^\top)^\dagger\|_F \|\Delta_i\|_F + 2\|(\widehat{W}^{(i)})^\dagger\|_F \right) \\
& + \|w_{M+1}^\star\|_2 \|(\widehat{W}^{(i)}(\widehat{W}^{(i)})^\top)^\dagger\|_F \|\widehat{\theta}_m^{(i)} - \theta_m^\star\|_2 \\
& + \|(\widehat{W}^{(i)}(\widehat{W}^{(i)})^\top)^\dagger\|_F \|M^{-1}\| \|\widehat{w}_m^{(i)}\|_2 \|\widehat{B}^{(i)^\top} X_{M+1}^{(i)^\top} Z_{M+1}^{(i)}\|_2
\end{aligned}
$$

Given matrix $\widehat{W}^{(i)} \in \mathbb{R}^{k \times M}$, utilizing the SVD, we can derive $\widehat{W}^{(i)}(\widehat{W}^{(i)})^\top = U\Sigma V^\top V \Sigma^\top U^\top = U(\Sigma\Sigma^\top)U^\top$. Consequently, we obtain $\|(\widehat{W}^{(i)}(\widehat{W}^{(i)})^\top)^\dagger\|_F = \|U(\Sigma\Sigma^\top)^\dagger U^\top\|_F = \sqrt{\sum_{i=1}^k (\frac{1}{\sigma_i^2})^2} \leqslant \frac{\sqrt{k}}{\sigma_{\min}^2(\widehat{W}^{(i)})}$ and $\|(\widehat{W}^{(i)})^\dagger\|_F = \sqrt{\sum_{i=1}^k \frac{1}{\sigma_i^2}} \leqslant \frac{\sqrt{k}}{\sigma_{\min}(\widehat{W}^{(i)})}$. According to Lemma B.1 in Lin et al. (2024), assume $SD(\widehat{B}^{(i)}, B^\star) \leqslant \delta^{(i)}$, with probability at least $1 - 2M \exp(k - cn_m^i)$, $\|M^{-1}\| \leqslant \frac{1}{0.9 n_{M+1}}$. According to Lemma B.2 in Lin et al. (2024), assume $\sigma^2 \leqslant \frac{k}{M} \delta^{(i)^2} \sigma_{\max}^{\star^2}$, and

$\mathrm{SD}(\widehat{B}^{(i)}, B^\star) \leqslant \delta^{(i)}$, if $\delta^{(i)} \leqslant \frac{0.02}{\mu\sqrt{k}\kappa}$, and if $n_m^i \geqslant C\max(\log M, \log d, k)$, then with probability at least $1 - 3\exp(\log M + k - cn_m^i)$,

$$\|\widehat{\theta}_m^{(i)} - \theta_m^\star\|_2 \leqslant 1.4\mu\delta^{(i)}\sqrt{\frac{k}{M}}\sigma_{\max}^\star,$$

$$\|\widehat{\Theta}^{(i)} - \Theta^\star\|_F \leqslant 1.4\mu\delta^{(i)}\sqrt{k}\sigma_{\max}^\star,$$

$$\sigma_{\min}(\widehat{W}^{(i)}) \geqslant 0.9\sigma_{\min}^\star,$$

$$\|\widehat{w}_m^{(i)}\| \leqslant 1.1\mu\sqrt{\frac{k}{M}}\sigma_{\max}^\star.$$

Based on Assumption 2.3, we have $\|w_m^\star\|^2 \leqslant \mu^2\frac{k}{M}\sigma_{\max}^{\star 2}$. In order to determine the upper bound for $\widehat{B}^{(i)^\top} X_{M+1}^{(i)^\top} Z_{M+1}^{(i)}$, let's consider a fixed $z \in \mathcal{S}_k$. we analyze $z^\top \widehat{B}^{(i)^\top} X_{M+1}^{(i)^\top} Z_{M+1}^{(i)} = \sum_{j=1}^{n_{M+1}} (\widehat{B}^{(i)} z)^\top x_{M+1,j}^{(i)} Z_{M+1,j}^{(i)}$, resulting in $\mathbb{E}[(\widehat{B}^{(i)} z)^\top x_{M+1,j}^{(i)} Z_{M+1,j}^{(i)}] = 0$ and

$$\begin{aligned}
\mathrm{Var}((\widehat{B}^{(i)} z)^\top x_{M+1,j}^{(i)}) &= \mathbb{E}[(\widehat{B}^{(i)} z)^\top x_{M+1,j}^{(i)}]^2 - (\mathbb{E}[(\widehat{B}^{(i)} z)^\top x_{M+1,j}^{(i)}])^2 \\
&= \mathbb{E}[(\widehat{B}^{(i)} z)^\top x_{M+1,j}^{(i)}]^2 \\
&= \mathbb{E}[z^\top \widehat{B}^{(i)^\top} x_{M+1,j}^{(i)} x_{M+1,j}^{(i)^\top} \widehat{B}^{(i)} z] \\
&= z^\top \widehat{B}^{(i)^\top} \mathbb{E}[x_{M+1,j}^{(i)} x_{M+1,j}^{(i)^\top}] \widehat{B}^{(i)} z \\
&= 1.
\end{aligned}$$

Given $Z_{M+1,j}^{(i)} \overset{\mathrm{iid}}{\sim} \mathcal{N}(0, \sigma^2)$, we have $\mathrm{Var}(Z_{M+1,j}^{(i)}) = \sigma^2$. Thus, $z^\top \widehat{B}^{(i)^\top} X_{M+1}^{(i)^\top} Z_{M+1}^{(i)}$ is a sum of $n_{M+1}$ subexponential random variables with parameter $K_j \leqslant C\sigma$. We apply the sub-exponential Bernstein inequality stated in Proposition A.1 by setting $g = Cn_{M+1}\sigma$. In order to implement this, we show that

$$\frac{g^2}{\sum_{j=1}^{n_{M+1}} K_j^2} \geqslant \frac{C^2 n_{M+1}^2 \sigma^2}{C^2 n_{M+1} \sigma^2} = n_{M+1}$$

$$\frac{g}{\max_j K_j} \geqslant \frac{Cn_{M+1}\sigma}{C\sigma} = n_{M+1}$$

Therefore, for a fixed $z \in \mathcal{S}_k$, with probability at least $1 - \exp(-cn_{M+1})$, $z^\top \widehat{B}^{(i)^\top} X_{M+1}^{(i)^\top} Z_{M+1}^{(i)} \leqslant Cn_{M+1}\sigma$. Using epsilon-net over all $z$ adds a factor of $\exp(k)$. Thus, with probability at least $1 - \exp(k - cn_{M+1})$, we have $\|\widehat{B}^{(i)^\top} X_{M+1}^{(i)^\top} Z_{M+1}^{(i)}\| \leqslant Cn_{M+1}\sigma \leqslant Cn_{M+1}\sqrt{\frac{k}{M}}\sigma_{\max}^\star \delta^{(i)}$. By combining the aforementioned results and the union bound, we can determine with probability at least $1 - \exp(d - \frac{cT\sum_{m=1}^M n_m^1}{\mu^6 \kappa^8 k^5}) - 5iTM\exp(k - cn_m^i) - \exp(k - cn_{M+1})$,

$$|\hat{\nu}_{i+1}(m)| - |\nu^\star(m)| \leqslant 3\mu^4\kappa^4\frac{k^3}{M}\delta^{(i)2} + 3.9\mu^3\kappa^3\frac{k^2\sqrt{k}}{M}\delta^{(i)} + 1.8\mu^2\kappa^2\frac{k\sqrt{k}}{\sqrt{M}}\delta^{(i)} + 1.5C\mu\kappa^2\frac{k\sqrt{k}}{M}\delta^{(i)}$$

$$\leqslant C\mu^4\kappa^4\frac{k^3}{M}\delta^{(i)} \leqslant \frac{0.02\epsilon_i}{M} \tag{18}$$

$$|\nu^\star(m)| - |\hat{\nu}_{i+1}(m)| \leqslant 3\mu^4\kappa^4\frac{k^3}{M}\delta^{(i)2} + 3.9\mu^3\kappa^3\frac{k^2\sqrt{k}}{M}\delta^{(i)} + 1.8\mu^2\kappa^2\frac{k\sqrt{k}}{\sqrt{M}}\delta^{(i)} + 1.5C\mu\kappa^2\frac{k\sqrt{k}}{M}\delta^{(i)} \tag{19}$$

$$\leqslant C\mu^4\kappa^4\frac{k^3}{M}\delta^{(i)} \leqslant \frac{0.02\epsilon_i}{M}$$

where equation 18 and equation 19 can be derived by setting $\delta_0^2 \leqslant \frac{0.02}{C\mu^4\kappa^4 k^3}$, $T \geqslant \frac{\log\frac{1}{\sqrt{2}}}{\log(1 - \frac{0.4\mu c\eta}{\kappa^2})}$ and applying Theorem 5.1 and Theorem 5.2 in (Lin et al., 2024). Following that, we can show with probability at least $1 - \exp(d - \frac{cT\sum_{m=1}^M n_m^1}{\mu^6 \kappa^8 k^5}) - 5iTM\exp(k - cn_m^i) - \exp(k - cn_{M+1})$,

$$|\nu^\star(m)| - \frac{0.02\epsilon_i}{M} \leqslant |\hat{\nu}_{i+1}(m)| \leqslant |\nu^\star(m)| + \frac{0.02\epsilon_i}{M}.$$

Hence, when considering $|\nu^\star(m)| \geqslant \frac{0.04\epsilon_i}{M}$, we can conclude

$$|\hat{\nu}_{i+1}(m)| \leqslant |\nu^\star(m)| + \frac{0.02\epsilon_i}{M} \leqslant \frac{3}{2}|\nu^\star(m)|$$

$$|\hat{\nu}_{i+1}(m)| \geqslant |\nu^\star(m)| - \frac{0.02\epsilon_i}{M} \geqslant \frac{1}{2}|\nu^\star(m)|.$$

When we consider $|\nu^\star(m)| \leqslant \frac{0.04\epsilon_i}{M}$, we can conclude

$$|\hat{\nu}_{i+1}(m)| \leqslant |\nu^\star(m)| + \frac{0.02\epsilon_i}{M} \leqslant \frac{0.06\epsilon_i}{M}$$

$$|\hat{\nu}_{i+1}(m)| \geqslant 0$$

Therefore, we prove with probability at least $1 - \exp(d - \frac{cT\sum_{m=1}^M n_m^1}{\mu^6\kappa^8 k^5}) - 5iTM\exp(k - cn_m^i) - \exp(k - cn_{M+1})$,

$$|\hat{\nu}_{i+1}(m)| = \begin{cases} [\frac{1}{2}|\nu^\star(m)|, \frac{3}{2}|\nu^\star(m)|], & \text{if } |\nu^\star(m)| \geqslant \frac{0.02\epsilon_i}{M} \\ [0, \frac{0.06\epsilon_i}{M}], & \text{if } |\nu^\star(m)| \leqslant \frac{0.02\epsilon_i}{M} \end{cases}$$

$\square$

**Lemma B.3.** *For any $\epsilon > 0$, success probabilities $\delta, \delta' \in [0, 1]$, let $\sigma^2 \leqslant \frac{c\|\theta_m^\star\|^2}{\mu k^3 \kappa^6 n_m^1}$, $\eta = \frac{0.4}{\sigma_{\max}^{\star 2}}$, and $T = C\kappa^2 \log\frac{1}{\epsilon}$. If*

$$\sum_{m=1}^M n_m^i \geqslant C\kappa^6\mu^2(d + M)k(\kappa^2 k^2 + \log\frac{1}{\epsilon})$$

*and*

$$n_m^i \geqslant C\max(\log d, \log M, k)\log\frac{1}{\epsilon},$$

*then after epoch $i$, with probability at least $1 - 2\delta - iTd^{-10} - 2de^{-\frac{\delta'^2 n_{M+1}}{3S}}$, it holds that*

$$\text{ER}(\widehat{B}, \widehat{w}_{M+1}) \leqslant \frac{\sigma^2(2k + 3\log\frac{1}{\delta})}{2(1 - \delta')n_{M+1}} + \frac{(1 + \delta')}{2(1 - \delta')^2}\mu^2 k \sigma_{\max}^{\star 2}\epsilon^2\left(2N(d - k) + 3\log\frac{1}{\delta}\right)\|\widetilde{\nu}^\star\|_2^2.$$

*where $\widetilde{\nu}^\star(m) = \frac{\nu^\star(m)}{\sqrt{n_m^i}}$, $\epsilon = \max(\epsilon', \epsilon_{noise})$, $\epsilon_{noise} = C\kappa^2\sqrt{NSR}$, $NSR := \frac{\sigma^2}{\min_m \|\theta_m^\star\|^2}$.*

*Proof.* From the definition of $\text{ER}(\widehat{B}, \widehat{w}_{M+1})$, we have

$$\text{ER}(\widehat{B}, \widehat{w}_{M+1}) = \frac{1}{2}\mathbb{E}_{x_{M+1,n}\sim p_{M+1}}\left[\left(x_{M+1,n}^\top(\widehat{B}\widehat{w}_{M+1} - B^\star w_{M+1}^\star)\right)^2\right]$$

$$= \frac{1}{2}(\widehat{B}\widehat{w}_{M+1} - B^\star w_{M+1}^\star)^\top(\widehat{B}\widehat{w}_{M+1} - B^\star w_{M+1}^\star) \tag{20}$$

$$\leqslant \frac{1}{2(1 - \delta')n_{M+1}}\|X_{M+1}(\widehat{B}\widehat{w}_{M+1} - B^\star w_{M+1}^\star)\|^2 \tag{21}$$

$$= \frac{1}{2(1 - \delta')n_{M+1}}\|X_{M+1}\widehat{B}((X_{M+1}\widehat{B})^\top(X_{M+1}\widehat{B}))^\dagger(X_{M+1}\widehat{B})^\top Y_{M+1} - X_{M+1}B^\star w_{M+1}^\star\|^2 \tag{22}$$

$$= \frac{1}{2(1 - \delta')n_{M+1}}\|P_{X_{M+1}\widehat{B}_T}(X_{M+1}B^\star w_{M+1}^\star + Z_{M+1}) - X_{M+1}B^\star w_{M+1}^\star\|^2$$

$$= \frac{1}{2(1 - \delta')n_{M+1}}\|P_{X_{M+1}\widehat{B}}Z_{M+1}\|^2 + \frac{1}{2(1 - \delta')n_{M+1}}\|P^\perp_{X_{M+1}\widehat{B}_T}X_{M+1}B^\star w_{M+1}^\star\|^2 \tag{23}$$

$$= \frac{1}{2(1 - \delta')n_{M+1}}\|P_{X_{M+1}\widehat{B}}Z_{M+1}\|^2 + \frac{1}{2(1 - \delta')n_{M+1}}\|P^\perp_{X_{M+1}\widehat{B}}X_{M+1}B^\star\widetilde{W}^\star\widetilde{\nu}^\star\|^2 \tag{24}$$

$$\leqslant \frac{1}{2(1 - \delta')n_{M+1}}\|P_{X_{M+1}\widehat{B}}Z_{M+1}\|^2 + \frac{1}{2(1 - \delta')n_{M+1}}\|P^\perp_{X_{M+1}\widehat{B}}X_{M+1}B^\star\widetilde{W}^\star\|_F^2\|\widetilde{\nu}^\star\|_2^2$$

where $\widetilde{W}^\star = W^\star\sqrt{\text{diag}([n_1, n_2, \cdots, n_M])}$ and $\widetilde{\nu}^\star(m) = \frac{\nu^\star(m)}{\sqrt{n_m}}$. equation 20 is derived from $\mathbb{E}\left[x_{M+1,n}x_{M+1,n}^\top\right] = I$. equation 21 is derived from Lemma C.1. equation 22 is derived from the least square estimator solution of the optimality of $\widehat{w}_{M+1}$. equation 23 is derived from $P^\perp_{X_{M+1}\widehat{B}}{}^\top P_{X_{M+1}\widehat{B}} = 0$. equation 24 is derived from $w^\star_{M+1} = \widetilde{W}^\star\widetilde{\nu}^\star$. Given that $Z_{M+1}$ follows i.i.d. Gaussian distribution with a zero mean and variance $\sigma^2$, it follows that $\frac{1}{\sigma^2}\|P_{X_{M+1}\widehat{B}}Z_{M+1}\|^2 \sim \chi^2(k)$. Applying the Chernoff bound for chi-square distribution, we have with probability at least $1 - \delta$, $\|P_{X_{M+1}\widehat{B}}Z_{M+1}\|^2 \leqslant \sigma^2(2k + 3\log\frac{1}{\delta})$. Following that, by combining the result obtained from Lemma C.2 along with applying the union bound, we derive that with probability at least $1 - 2\delta - iTd^{-10} - 2de^{-\frac{\delta'^2 n_{M+1}}{3S}}$,

$$\text{ER}(\widehat{B}, \widehat{w}_{M+1}) \leqslant \frac{\sigma^2(2k + 3\log\frac{1}{\delta})}{2(1-\delta')n_{M+1}} + \frac{(1+\delta')}{2(1-\delta')^2}\mu^2 k\sigma^{\star^2}_{\max}\epsilon^2\left(2N(d-k) + 3\log\frac{1}{\delta}\right)\|\widetilde{\nu}^\star\|_2^2.$$

$\square$

**Theorem B.4.** *For any $\epsilon > 0$, success probabilities $\delta, \delta' \in [0,1]$, let $\sigma^2 \leqslant \frac{c\|\theta^\star_m\|^2}{\mu k^3\kappa^6 n^1_m}$, $\eta = \frac{0.4}{\sigma^{\star^2}_{\max}}$, and $T = C\kappa^2\log\frac{1}{\epsilon}$. If*

$$\sum_{m=1}^M n^i_m \geqslant C\kappa^6\mu^2(d+M)k(\kappa^2 k^2 + \log\frac{1}{\epsilon})$$

*and*

$$n^i_m \geqslant C\max(\log d, \log M, k)\log\frac{1}{\epsilon},$$

*then after epoch $i$, with probability at least $1 - 2\delta - iTd^{-10} - 2de^{-\frac{\delta'^2 n_{M+1}}{3S}} - \exp(k - cn_{M+1})$, it holds that*

$$\text{ER}(\widehat{B}, \widehat{w}_{M+1}) \leqslant \frac{\sigma^2(2k + 3\log\frac{1}{\delta})}{2(1-\delta')n_{M+1}} + \frac{2(1+\delta')\epsilon_i^2}{\beta(1-\delta')^2}\mu^2 k\sigma^{\star^2}_{\max}\epsilon^2\left(2\sum_{m=1}^M n^i_m(d-k) + 3\log\frac{1}{\delta}\right)s^\star_i,$$

*where $\|\nu^\star\|^i_{0,\gamma} := \left|\left\{m : |\nu^\star(m)| > \sqrt{\gamma\frac{\|\nu^\star\|_2^2}{\sum_{i=1}^M n^i_m}}\right\}\right| = \left|\left\{m : |\nu^\star(m)| > 0.02\sqrt{\gamma\frac{\epsilon_i}{M}}\right\}\right|$ by setting $\beta\epsilon_i^{-2} = \frac{\sum_{m=1}^M n^i_m}{\|\nu^\star\|_2^2}$ with $\beta = 2500M^2$, $s^\star_i = (1-\gamma)\|\nu^\star\|^i_{0,\gamma} + \gamma M$, $\epsilon = \max(\epsilon', \epsilon_{noise})$, $\epsilon_{noise} = C\kappa^2\sqrt{NSR}$, $NSR := \frac{\sigma^2}{\min_m\|\theta^\star_m\|^2}$.*

*Proof.* From Lemma B.3, we can obtain that with probability at least $1 - 2\delta - iTd^{-10} - 2de^{-\frac{\delta'^2 n_m}{3S}}$,

$$\text{ER}(\widehat{B}, \widehat{w}_{M+1}) \leqslant \frac{\sigma^2(2k + 3\log\frac{1}{\delta})}{2(1-\delta')n_{M+1}} + \frac{(1+\delta')}{2(1-\delta')^2}\mu^2 k\sigma^{\star^2}_{\max}\epsilon^2\left(2\sum_{m=1}^M n^i_m(d-k) + 3\log\frac{1}{\delta}\right)\|\widetilde{\nu}^\star\|_2^2,$$

where $\widetilde{\nu}^\star(m) = \frac{\nu^\star(m)}{\sqrt{n_m^i}}$. Furthermore, for any $\gamma \in [0,1]$, with probability at least $1 - \exp(d - \frac{cT \sum_{m=1}^{M} n_m^1}{\mu^6 \kappa^8 k^5}) - 5iTM \exp(k - c\min\{n_m^i\}) - \exp(k - cn_{M+1})$, we have

$$\sum_{m=1}^{M} \frac{\nu^\star(m)^2}{n_m^i} = \sum_{m=1}^{M} \frac{\nu^\star(m)^2}{n_m^i} \left( \mathbb{1}\{|\nu^\star(m)| \leqslant 0.02\sqrt{\gamma}\frac{\epsilon_i}{M}\} + \mathbb{1}\{0.02\sqrt{\gamma}\frac{\epsilon_i}{M} \leqslant |\nu^\star(m)| \leqslant 0.02\frac{\epsilon_i}{M}\} \right)$$

$$+ \sum_{m=1}^{M} \frac{\nu^\star(m)^2 \epsilon_i^2}{\beta \hat{\nu}(m)^2} \mathbb{1}\{|\nu^\star(m)| \geqslant 0.02\frac{\epsilon_i}{M}\} \tag{25}$$

$$\leqslant \sum_{m=1}^{M} 0.02^2 \gamma \frac{\epsilon_i^2}{M^2} \left( \mathbb{1}\{|\nu^\star(m)| \leqslant 0.02\sqrt{\gamma}\frac{\epsilon_i}{M}\} + \mathbb{1}\{0.02\sqrt{\gamma}\frac{\epsilon_i}{M} \leqslant |\nu^\star(m)| \leqslant 0.02\frac{\epsilon_i}{M}\} \right)$$

$$+ \frac{\epsilon_i^2}{\beta} \sum_{m=1}^{M} \frac{4\hat{\nu}(m)^2}{\hat{\nu}(m)^2} \mathbb{1}\{|\nu^\star(m)| \geqslant 0.02\frac{\epsilon_i}{M}\} \tag{26}$$

$$\leqslant \frac{4\epsilon_i^2}{\beta} \sum_{m=1}^{M} \left( \gamma \mathbb{1}\{|\nu^\star(m)| \leqslant 0.02\sqrt{\gamma}\frac{\epsilon_i}{M}\} + \mathbb{1}\{0.02\sqrt{\gamma}\frac{\epsilon_i}{M} \leqslant |\nu^\star(m)| \leqslant 0.02\frac{\epsilon_i}{M}\} + \mathbb{1}\{|\nu^\star(m)| \geqslant 0.02\frac{\epsilon_i}{M}\} \right)$$

$$\tag{27}$$

$$= \frac{4\epsilon_i^2}{\beta} \sum_{m=1}^{M} \left( \gamma \mathbb{1}\{|\nu^\star(m)| \leqslant 0.02\sqrt{\gamma}\frac{\epsilon_i}{M}\} + \mathbb{1}\{|\nu^\star(m)| \geqslant 0.02\sqrt{\gamma}\frac{\epsilon_i}{M}\} \right)$$

$$\leqslant \frac{4\epsilon_i^2}{\beta} \left( \gamma(M - \|\nu^\star\|_{0,\gamma}^i) + \|\nu^\star\|_{0,\gamma}^i \right)$$

$$= \frac{4\epsilon_i^2}{\beta} \left( (1-\gamma)\|\nu^\star\|_{0,\gamma}^i + \gamma M \right)$$

where equation 25 can be derived from $n_m^i = \beta \hat{\nu}(m)^2 \epsilon_i^{-2}$. equation 26 can be obtained from Lemma B.2. equation 27 can be derived from $\beta = 2500M^2$. By using the union bound, we conclude that with probability at least $1 - 2\delta - iTd^{-10} - 2de^{-\frac{\delta'^2 n_{M+1}}{3S}} - \exp(k - cn_{M+1})$,

$$\mathrm{ER}(\widehat{B}, \widehat{w}_{M+1}) \leqslant \frac{\sigma^2(2k + 3\log\frac{1}{\delta})}{2(1-\delta')n_{M+1}} + \frac{2(1+\delta')\epsilon_i^2}{\beta(1-\delta')^2}\mu^2 k \sigma_{\max}^{\star 2} \epsilon^2 \left( 2\sum_{m=1}^{M} n_m^i(d-k) + 3\log\frac{1}{\delta} \right) s_i^\star.$$

To ensure the validity of the results from Lin et al. (2024), it is required that

$$\sum_{m=1}^{M} n_m^i \geqslant C\kappa^6 \mu^2 (d+M)k(\kappa^2 k^2 + \log\frac{1}{\epsilon})$$

and

$$n_m^i \geqslant C\max(\log d, \log M, k)\log\frac{1}{\epsilon}.$$

$\square$

**Lemma B.5.** *Assume* $\sigma^2 \leqslant \frac{k}{M}\delta^{(i)^2}\sigma_{\max}^{\star 2}$. *Set* $\eta = \frac{0.4}{\sigma_{\max}^{\star 2}}$, $\delta_0^2 \leqslant \frac{0.02}{C\mu^4 \kappa^4 k^3}$ *and* $T \geqslant \frac{\log\frac{1}{\sqrt{2}}}{\log(1 - \frac{0.4\mu c_\eta}{\kappa^2})}$. *If*

$$\sum_{i=1}^{M} n_m^i \geqslant C\kappa^4 \mu^2 dk \text{ and } n_m^i \gtrsim \max(\log d, \log M, k),$$

*then with probability at least* $1 - \exp(d - \frac{cT \sum_{m=1}^{M} n_m^1}{\mu^6 \kappa^8 k^5}) - 5iTM \exp(k - c\min\{n_m^i\}) - \exp(k - c\epsilon_i n_{M+1})$, *the total number of source samples for any epoch* $i$ *is determined by*

$$O(\beta(\epsilon_i^{-2}\|\nu^\star\|_2^2 + \frac{0.06^2}{M})).$$

*Proof.* Considering any given epoch $i$, we can derive

$$\sum_{m=1}^{M} n_m^i = \sum_{m=1}^{M} \beta \hat{\nu}(m)^2 \epsilon_i^{-2} \tag{28}$$

$$= \sum_{m=1}^{M} \beta \hat{\nu}(m)^2 \epsilon_i^{-2} \mathbb{1}\{|\nu^\star| > \frac{0.02\epsilon_i}{M}\} + \sum_{m=1}^{M} \beta \hat{\nu}(m)^2 \epsilon_i^{-2} \mathbb{1}\{|\nu^\star| \leqslant \frac{0.02\epsilon_i}{M}\}$$

$$\leqslant \frac{9}{4} \sum_{m=1}^{M} \beta \nu^\star(m)^2 \epsilon_i^{-2} \mathbb{1}\{|\nu^\star| > \frac{0.02\epsilon_i}{M}\} + \sum_{m=1}^{M} \beta \frac{0.06^2}{M^2} \mathbb{1}\{|\nu^\star| \leqslant \frac{0.02\epsilon_i}{M}\} \tag{29}$$

$$\leqslant \frac{9}{4} \sum_{m=1}^{M} \beta \nu^\star(m)^2 \epsilon_i^{-2} + \sum_{m=1}^{M} \beta \frac{0.06^2}{M^2}$$

$$= \frac{9}{4} \beta \epsilon_i^{-2} \|\nu^\star\|_2^2 + \beta \frac{0.06^2}{M}$$

$$= O(\beta(\epsilon_i^{-2}\|\nu^\star\|_2^2 + \frac{0.06^2}{M})),$$

where equation 28 is derived from $n_m^i = \beta \hat{\nu}(m)^2 \epsilon_i^{-2}$. equation 29 is derived from Lemma B.2. $\square$

## B.2 PROOF OF THEOREM 5.1

*Proof.* From Theorem B.4, under the given assumptions and conditions, with probability at least $1 - 2\delta - \Gamma T d^{-10} - 2de^{-\frac{\delta'^2 n_{M+1}}{3S}} - \exp(k - cn_{M+1})$, the excess risk at the end of the last epoch $\Gamma$ is determined by

$$\mathrm{ER}(\widehat{B}, \widehat{w}_{M+1}) \leqslant \frac{\sigma^2(2k + 3\log\frac{1}{\delta})}{2(1-\delta')n_{M+1}} + \frac{2(1+\delta')\epsilon_\Gamma^2}{\beta(1-\delta')^2}\mu^2 k \sigma_{\max}^{\star 2} \epsilon^2 \left(2\sum_{m=1}^{M} n_m^i(d-k) + 3\log\frac{1}{\delta}\right) s_\Gamma^\star.$$

Based on Lemma B.5, under the given assumptions and conditions, with probability at least $1 - \exp(d - \frac{cT\sum_{m=1}^{M} n_m^1}{\mu^6 \kappa^8 k^5}) - 5iTM\exp(k - c\min\{n_m^\Gamma\}) - \exp(k - cn_{M+1})$, we have

$$N = \sum_{i=1}^{\Gamma}\sum_{m=1}^{M} n_m^i$$

$$\leqslant \sum_{i=1}^{\Gamma} \left(\frac{9}{4}\beta\epsilon_i^{-2}\|\nu^\star\|_2^2 + \beta\frac{0.06^2}{M}\right)$$

$$\leqslant \sum_{i=1}^{\Gamma} \left(\frac{9}{4}\beta\epsilon_i^{-2}\|\nu^\star\|_2^2 + \beta\frac{0.06^2}{M}\epsilon_i^{-2}\right)$$

$$\leqslant \left(3\|\nu^\star\|_2^2 + \frac{0.0048}{M}\right)\beta\epsilon_\Gamma^{-2}$$

Define $t := 6(1+\delta')\mu^2 k\sigma_{\max}^{\star 2} s_\Gamma^\star(\|\nu^\star\|_2^2 + \frac{0.0016}{M})$. For $C' > 1$, setting source sample size $N \geqslant \frac{3C'}{c(1-\delta')^2} t\epsilon\log\frac{1}{\delta}$ results in

$$N \geqslant \frac{3C'}{c(1-\delta')^2} t\epsilon\log\frac{1}{\delta}$$

$$= \frac{3\log\frac{1}{\delta}}{2(d-k)}C'\left(\frac{2}{c(1-\delta')^2}(d-k)t\epsilon\right)$$

$$\geqslant \frac{3\log\frac{1}{\delta}}{2(d-k)}\frac{\frac{2}{c(1-\delta')^2}(d-k)t\epsilon}{1 - \frac{2}{c(1-\delta')^2}(d-k)t\epsilon} \tag{30}$$

$$= \frac{3t\log\frac{1}{\delta}}{c(1-\delta')^2\epsilon^{-1} - 2(d-k)t}$$

where equation 30 is derived from the fact that there exists a constant $C' > 1$ satisfying the inequality $\frac{x}{1-x} \leqslant C'x$ for $0 < x < 1$. Consequently, we find that $\frac{1}{(1-\delta')^2}\left(2(d-k) + \frac{3}{N}\log\frac{1}{\delta}\right)t\epsilon^2 \leqslant c\epsilon$. Using the bound $N \leqslant \left(3\|\nu^\star\|_2^2 + \frac{0.0048}{M}\right)\beta\epsilon_\Gamma^{-2}$, we derive

$$\frac{1}{(1-\delta')^2}\left(2(d-k) + \frac{3}{N}\log\frac{1}{\delta}\right)t\epsilon^2 = \frac{1}{(1-\delta')^2}\left(2N(d-k) + 3\log\frac{1}{\delta}\right)\frac{t\epsilon^2}{N}$$

$$\geqslant \frac{1}{(1-\delta')^2}\left(2N(d-k) + 3\log\frac{1}{\delta}\right)\frac{t\epsilon^2}{\left(3\|\nu^\star\|_2^2 + \frac{0.0048}{M}\right)\beta\epsilon_\Gamma^{-2}}$$

$$= \frac{2(1+\delta')\epsilon_\Gamma^2}{\beta(1-\delta')^2}\mu^2 k\sigma_{\max}^{\star 2}\epsilon^2\left(2N(d-k) + 3\log\frac{1}{\delta}\right)s_\Gamma^\star$$

$$\geqslant \frac{2(1+\delta')\epsilon_\Gamma^2}{\beta(1-\delta')^2}\mu^2 k\sigma_{\max}^{\star 2}\epsilon^2\left(2\sum_{m=1}^{M}n_m^i(d-k) + 3\log\frac{1}{\delta}\right)s_\Gamma^\star$$

Thus, we conclude $\frac{2(1+\delta')\epsilon_\Gamma^2}{\beta(1-\delta')^2}\mu^2 k\sigma_{\max}^{\star 2}\epsilon^2\left(2\sum_{m=1}^{M}n_m^i(d-k) + 3\log\frac{1}{\delta}\right)s_\Gamma^\star \leqslant c\epsilon$. By setting the target sample size as

$$n_{M+1} \geqslant \frac{\sigma^2(2k + 3\log\frac{1}{\delta})\epsilon^{-1}}{2(1-c)(1-\delta')},$$

where for $0 < c < 1$, we derive

$$\frac{\sigma^2(2k + 3\log\frac{1}{\delta})}{2(1-\delta')n_{M+1}} \leqslant (1-c)\epsilon.$$

Based on this analysis, we can conclude that with probability at least $1 - 2\delta - \Gamma T d^{-10} - 2de^{-\frac{\delta'^2 n_{M+1}}{3S}} - \exp(-cn_{M+1})$,

$$\mathrm{ER}(\widehat{B}, \widehat{w}_{M+1}) \leqslant \epsilon.$$

$\square$

## C  Adaptive Sampling for Low-Rank Representation Learning for $\nu^\star$ Known Setting

### C.1  Adaptive Sampling Algorithm for LRRL ($\nu^\star$ Known Setting)

In this section, we present the alternating GD and minimization estimator-based adaptive sampling algorithm when the relevance parameter $\nu^\star$ is known. The key difference with respect to Algorithm 1 is that after estimating $\Theta^\star$ and $\widehat{w}_{M+1}$ we can assign the number of samples for the next epoch using the true values $\propto \nu^\star(m)$. The proof approach is simplified as compared to that of the unknown setting since $\nu^\star$ is known. We present the pseudocode of the algorithm in Algorithm 2. We note that Algorithm 2 is an iterative algorithm with one epoch (i.e., $\Gamma = 1$).

We first present the complete statement of Theorem 5.2.

**Complete form of Theorem 5.2:** Consider Assumptions 2.2 and 2.3 hold. For any $\epsilon' > 0$, $\delta, \delta' \in [0, 1]$, $0 < c < 1$, $C > 1$, let $\sigma^2 \leqslant \frac{c\|\theta_m^\star\|^2}{\mu k^3 \kappa^6 n_m}$, $\eta = \frac{0.4}{\sigma_{\max}^{\star 2}}$, and $T = C\kappa^2\log\frac{1}{\epsilon}$. If $n_m \geqslant C\max(\log d, \log M, k)\log\frac{1}{\epsilon}$, then with probability $1 - 2\delta - Td^{-10} - 2de^{-\frac{\delta'^2 n_{M+1}}{3S}}$, the output of Algorithm 2 guarantees that

$$\mathrm{ER}(\widehat{B}_T, \widehat{w}_{M+1}) \leqslant \epsilon,$$

whenever the total sampling budget from all sources $N$ is at least

$$O\left(\min\left\{\frac{(1+\delta')}{(1-\delta')^2}k\|\nu^\star\|_2^2 s^\star\epsilon\log\frac{1}{\delta}, (d+M)k(k^2 + \log\frac{1}{\epsilon})\right\}\right)$$

and the number of target samples $n_{M+1}$ is at least

$$n_{M+1} \geqslant \frac{\sigma^2(2k + 3\log\frac{1}{\delta})}{2(1-c)(1-\delta')}\epsilon^{-1},$$

---

**Algorithm 2**: Active Low-Rank Representation Learning Algorithm for Known $\nu^\star$ (A-LRRL-known)

---

1: **Input:** Confidence $\delta$, representation function class $\Phi$, relevance parameter $\nu^\star$, source-task sampling budget $N \gg M(\frac{k}{\sqrt{M^3}}((d-k)+\log(\frac{1}{\delta})))$, multiplier for $\alpha$ in init step, $\tilde{C}$, GD step size $\eta$, number of GD iterations $T$

2: Initialize the lower bound $\underline{N} = \frac{k}{\sqrt{M^3}}((d-k)+\log(\frac{1}{\delta}))$ and number of samples $n_m = \max\{(N-M\underline{N})\frac{(\nu^\star(m))^2}{\|\nu^\star\|_2^2}, \underline{N}\}$

3: For each task $m$, draw $n_m$ i.i.d samples from the corresponding offline dataset denoted as $\{X_m, Y_m\}_{m=1}^M$

4: Set $\alpha = \frac{\tilde{C}}{NM}\sum_{m=1,n=1}^{M,n_m} y_{m,n}^2$

5: $y_{m,trunc}(\alpha) := Y_m \circ \mathbb{1}_{\{|Y_m| \leqslant \sqrt{\alpha}\}}$

6: $\widehat{\Theta}_0 := \sum_{m=1}^M \frac{1}{n_m} X_m^\top y_{m,trunc}(\alpha)e_m^\top$

7: Set $\widehat{B}_0 \leftarrow$ top-$k$-singular-vectors of $\widehat{\Theta}_0$

8: **GDmin iterations:**

9: **for** $\ell = 1$ to $T$ **do**

10:     Let $\widehat{B} \leftarrow \widehat{B}_{\ell-1}$

11:     **Update** $\widehat{w}_m, \widehat{\theta}_m$**:** For each $m \in [M]$, set $(\widehat{w}_m)_\ell \leftarrow (X_m\widehat{B})^\dagger Y_m$ and set $(\widehat{\theta}_m)_\ell \leftarrow \widehat{B}(\widehat{w}_m)_\ell$

12:     **Gradient w.r.t** $\widehat{B}$**:** Compute $\nabla_{\widehat{B}}f(\widehat{B},\widehat{W}_\ell) = \sum_{m=1}^M X_m^\top(X_m\widehat{B}(\widehat{w}_m)_\ell - Y_m)(\widehat{w}_m)_\ell^\top$

13:     **GD step:** Set $\widehat{B}^+ \leftarrow \widehat{B} - \frac{\eta}{N/M}\nabla_{\widehat{B}}f(\widehat{B},\widehat{W}_\ell)$

14:     **Projection step:** Compute $\widehat{B}^+ \overset{QR}{=} B^+R^+$

15:     Set $\widehat{B}_\ell \leftarrow B^+$

16: **end for**

17: Observe $n_{M+1}$ samples $X_{M+1}$ and $Y_{M+1}$ for the target task

18: Compute $\widehat{w}_{M+1} = \arg\min_w \|X_{M+1}^\top\widehat{B}_T w - Y_{M+1}\|^2$

19: Return $\widehat{B}_T, \widehat{w}_{M+1}$

---

where $s^\star = (1-\gamma)\|\nu^\star\|_{0,\gamma} + \gamma M$, $\|\nu^\star\|_{0,\gamma} := \left|\left\{m : |\nu_m^\star| > \sqrt{\gamma\frac{\|\nu^\star\|_2^2}{N}}\right\}\right|$ for $\gamma \in [0,1]$, $\epsilon = \max(\epsilon', \epsilon_{noise})$, $\epsilon_{noise} = C\kappa^2\sqrt{NSR}$, $NSR := \frac{\sigma^2}{\min_m \|\theta_m^\star\|^2}$.

## C.2 Supporting Results and Proofs for the $\nu^\star$ Known Setting

We present initial lemmas and then prove our main theorem.

**Lemma C.1.** *Assume $\|x_{m,n}\|^2 \leqslant S$ for any $m \in [M+1]$ and $n \in [n_m]$. For any $m \in [M+1]$, with probability at least $1 - 2de^{-\frac{\delta'^2 n_m}{3S}}$, it holds that $(1-\delta')n_m I \preceq X_m^\top X_m \preceq (1+\delta')n_m I$, where $n_m$ denotes the number of rows in $X_m$.*

*Proof.* Given that $X_m^\top X_m = \sum_{n=1}^{n_m} x_{m,n}x_{m,n}^\top$, where $x_{m,n}x_{m,n}^\top \succeq 0$ and $\lambda_{\max}(x_{m,n}x_{m,n}^\top) \leqslant S$. Since

$$\lambda_{\min}(\sum_{n=1}^{n_m}\mathbb{E}[x_{m,n}x_{m,n}^\top]) = \lambda_{\max}(\sum_{n=1}^{n_m}\mathbb{E}[x_{m,n}x_{m,n}^\top]) = n_m,$$

by applying the Matrix Chernoff inequality, we have with probability at least $1 - de^{-\frac{\delta'^2 n_m}{2S}}$, $\lambda_{\min}(\sum_{n=1}^{n_m} x_{m,n}x_{m,n}^\top) \geqslant (1-\delta')n_m$ and with probability at least $1 - de^{-\frac{\delta'^2 n_m}{3S}}$, $\lambda_{\max}(\sum_{n=1}^{n_m} x_{m,n}x_{m,n}^\top) \leqslant (1+\delta')n_m$. Applying union bound completes the proof. $\square$

Define $P_A := A(A^\top A)^\dagger A^\top$ and $P_A^\perp = I - I_A$.

**Lemma C.2.** *Assume that Assumptions 2.2 and 2.3 hold and $\sigma^2 \leqslant \frac{c\|\theta_m^\star\|^2}{\mu k^3 \kappa^6 n_m}$. Set $\eta = \frac{0.4}{\sigma_{\max}^{\star 2}}$ and $T = C\kappa^2 \log\frac{1}{\epsilon}$. If $N \geqslant C\kappa^6\mu^2(d+M)k(\kappa^2 k^2 + \log\frac{1}{\epsilon})$ and $n_m \geqslant C\max(\log d, \log M, k)\log\frac{1}{\epsilon}$,*

*then with probability at least $1 - \delta - Td^{-10} - 2de^{-\frac{\delta'^2 n_{M+1}}{3S}}$,*

$$\frac{1}{n_{M+1}}\|P^{\perp}_{X_{M+1}\widehat{B}_T}X_{M+1}B^{\star}\widetilde{W}^{\star}\|_F^2 \leqslant \frac{(1+\delta')}{(1-\delta')}\epsilon^2\mu^2 k\sigma^{\star}_{\max}{}^2\left(2N(d-k) + 3\log\frac{1}{\delta}\right),$$

*where $\widetilde{W}^{\star} = W^{\star}\sqrt{\operatorname{diag}([n_1, n_2, \cdots, n_M])}$, $\epsilon = \max(\epsilon', \epsilon_{noise})$, $\epsilon_{noise} = C\kappa^2\sqrt{NSR}$, $NSR := \frac{\sigma^2}{\min_m \|\theta^{\star}_m\|^2}$.*

*Proof.* Given two matrices $A_1$ and $A_2$ with the same number of columns that satisfy $A_1^{\top}A_1 \succeq A_2^{\top}A_2$, for any two matrices $B$ and $B'$ with compatible dimensions, from Lemma A.7 from Du et al. (2020), we have the following inequality

$$\|P^{\perp}_{A_1 B}A_1 B'\|_F^2 \geqslant \|P^{\perp}_{A_2 B}A_2 B'\|_F^2.$$

Using the above result and Lemma C.1, with probability at least $1 - 2de^{-\frac{\delta'^2 n_{M+1}}{3S}}$, the following inequalities hold.

$$\frac{1}{n_{M+1}}\|P^{\perp}_{X_{M+1}\widehat{B}_T}X_{M+1}B^{\star}\widetilde{W}^{\star}\|_F^2 \leqslant (1+\delta')\|P^{\perp}_{I\widehat{B}_T}IB^{\star}\widetilde{W}^{\star}\|_F^2 \leqslant \frac{(1+\delta')}{(1-\delta')}\sum_{m=1}^M \|P^{\perp}_{X_m\widehat{B}_T}X_m B^{\star}w^{\star}_m\|_2^2. \tag{31}$$

Using the definition of $P^{\perp}_{X_m\widehat{B}_T}X_m B^{\star}w^{\star}_m$, where $\widehat{B}_T$ is the estimate in the $T$-th GD iteration, we have

$$\sum_{m=1}^M \|P^{\perp}_{X_m\widehat{B}_T}X_m B^{\star}w^{\star}_m\|_2^2 = \sum_{m=1}^M \|X_m B^{\star}w^{\star}_m - (X_m\widehat{B}_T)((X_m\widehat{B}_T)^{\top}(X_m\widehat{B}_T))^{-1}(X_m\widehat{B}_T)^{\top}X_m B^{\star}w^{\star}_m\|_2^2$$

$$= \sum_{m=1}^M \|X_m(B^{\star}w^{\star}_m - \widehat{B}_T(\widehat{w}_m)_T) - (X_m\widehat{B}_T)((X_m\widehat{B}_T)^{\top}(X_m\widehat{B}_T))^{-1}$$
$$(X_m\widehat{B}_T)^{\top}X_m(B^{\star}w^{\star}_m - \widehat{B}_T(\widehat{w}_m)_T)\|_2^2 \tag{32}$$

$$= \sum_{m=1}^M \|P^{\perp}_{X_m\widehat{B}_T}X_m(B^{\star}w^{\star}_m - \widehat{B}_T(\widehat{w}_m)_T)\|_2^2$$

$$\leqslant \left(\sum_{m=1}^M \|P^{\perp}_{X_m\widehat{B}_T}X_m\|_F^2\right)\cdot\left(\sum_{m=1}^M \|B^{\star}w^{\star}_m - \widehat{B}_T(\widehat{w}_m)_T\|_2^2\right) \tag{33}$$

$$= \|B^{\star}W^{\star} - \widehat{B}_T\widehat{W}_T\|_F^2 \sum_{m=1}^M \|P^{\perp}_{X_m\widehat{B}_T}X_m\|_F^2. \tag{34}$$

equation 32 is derived from adding and subtracting and by using $X_m\widehat{B}_T(\widehat{w}_m)_T - (X_m\widehat{B}_T)((X_m\widehat{B}_T)^{\top}(X_m\widehat{B}_T))^{-1}(X_m\widehat{B}_T)^{\top}X_m\widehat{B}_T(\widehat{w}_m)_T = X_m\widehat{B}_T(\widehat{w}_m)_T - X_m\widehat{B}_T(\widehat{w}_m)_T = 0$. equation 33 is derived from Cauchy-Schwarz inequality. Given that $X_m$ follows i.i.d. standard Gaussian distribution, it follows that $\sum_{m=1}^M \|P^{\perp}_{X_m\widehat{B}_T}X_m\|_F^2 \sim \chi^2(\sum_{m=1}^M n_m(d-k))$. Applying the Chernoff bound for chi-square distribution, we have

$$\sum_{m=1}^M \|P^{\perp}_{X_m\widehat{B}_T}X_m\|_F^2 \leqslant \sum_{m=1}^M n_m(d-k) + 2\sqrt{\sum_{m=1}^M n_m(d-k)\log\frac{1}{\delta}} + 2\log\frac{1}{\delta},$$

with probability at least $1 - \delta$. Using the inequality $\sqrt{ab} \leqslant \frac{a+b}{2}$, we can determine $2\sqrt{\sum_{m=1}^M n_m(d-k)\log\frac{1}{\delta}} \leqslant \sum_{m=1}^M n_m(d-k) + \log\frac{1}{\delta}$. Therefore, we conclude that with probability at least $1 - \delta$, $\sum_{m=1}^M \|P^{\perp}_{X_m\widehat{B}_T}X_m\|_F^2 \leqslant 2\sum_{m=1}^M n_m(d-k) + 3\log\frac{1}{\delta}$. From Theorem 5.3 in Lin et al. (2024), under the given assumptions and conditions, with probability at least $1 - Td^{-10}$, $\|\widehat{\theta}_{m,T} - \theta^{\star}_m\| \leqslant \epsilon\|\theta^{\star}_m\|$ for all $m \in [M]$. Then we have with probability at least $1 - Td^{-10}$,

$$\|\widehat{B}_T\widehat{W}_T - B^{\star}W^{\star}\|_F^2 \leqslant \sum_{m=1}^M \epsilon^2\|\theta^{\star}_m\|^2 \leqslant \epsilon^2\mu^2 k\sigma^{\star}_{\max}{}^2.$$

The above inequality uses the fact that $B^\star$ is a unitary matrix and Assumption 2.3. Hence, by combining these results and using the union bound, we conclude that with probability at least $1 - \delta - Td^{-10} - 2de^{-\frac{\delta'^2 n_{M+1}}{3S}}$, we have

$$\sum_{m=1}^{M} \|P_{X_m \widehat{B}_T}^\perp X_m B^\star w_m^\star\|_2^2 \leqslant \frac{(1+\delta')}{(1-\delta')} \|B^\star W^\star - \widehat{B}_T \widehat{W}_T\|_F^2 \sum_{m=1}^{M} \|P_{X_m \widehat{B}_T}^\perp X_m\|_F^2$$

$$\leqslant \epsilon^2 \mu^2 k {\sigma_{\max}^\star}^2 \left( 2 \sum_{m=1}^{M} n_m(d-k) + 3\log\frac{1}{\delta} \right).$$

Substituting in equation 31 completes the proof. $\qquad\square$

### C.3  PROOF OF THEOREM 5.2

*Proof.* From the definition of $\mathrm{ER}(\widehat{B}_T, \widehat{w}_{M+1})$, we have

$$\mathrm{ER}(\widehat{B}_T, \widehat{w}_{M+1})$$

$$= \frac{1}{2}\mathbb{E}_{x_{M+1,n} \sim p_{M+1}} \left[ \left( x_{M+1,n}^\top (\widehat{B}_T \widehat{w}_{M+1} - B^\star w_{M+1}^\star) \right)^2 \right]$$

$$= \frac{1}{2}(\widehat{B}_T \widehat{w}_{M+1} - B^\star w_{M+1}^\star)^\top (\widehat{B}_T \widehat{w}_{M+1} - B^\star w_{M+1}^\star) \tag{35}$$

$$\leqslant \frac{1}{2(1-\delta')n_{M+1}} \|X_{M+1}(\widehat{B}_T \widehat{w}_{M+1} - B^\star w_{M+1}^\star)\|^2 \tag{36}$$

$$= \frac{1}{2(1-\delta')n_{M+1}} \|X_{M+1}\widehat{B}_T((X_{M+1}\widehat{B}_T)^\top (X_{M+1}\widehat{B}_T))^\dagger$$

$$(X_{M+1}\widehat{B}_T)^\top Y_{M+1} - X_{M+1}B^\star w_{M+1}^\star\|^2 \tag{37}$$

$$= \frac{1}{2(1-\delta')n_{M+1}} \|P_{X_{M+1}\widehat{B}_T}(X_{M+1}B^\star w_{M+1}^\star + Z_{M+1})$$

$$- X_{M+1}B^\star w_{M+1}^\star\|^2$$

$$= \frac{1}{2(1-\delta')n_{M+1}} \|P_{X_{M+1}\widehat{B}_T} Z_{M+1}\|^2$$

$$+ \frac{1}{2(1-\delta')n_{M+1}} \|P_{X_{M+1}\widehat{B}_T}^\perp X_{M+1}B^\star w_{M+1}^\star\|^2 \tag{38}$$

$$= \frac{1}{2(1-\delta')n_{M+1}} \|P_{X_{M+1}\widehat{B}_T} Z_{M+1}\|^2$$

$$+ \frac{1}{2(1-\delta')n_{M+1}} \|P_{X_{M+1}\widehat{B}_T}^\perp X_{M+1}B^\star \widetilde{W}^\star \widetilde{\nu}^\star\|^2 \tag{39}$$

$$\leqslant \frac{1}{2(1-\delta')n_{M+1}} \|P_{X_{M+1}\widehat{B}_T} Z_{M+1}\|^2$$

$$+ \frac{1}{2(1-\delta')n_{M+1}} \|P_{X_{M+1}\widehat{B}_T}^\perp X_{M+1}B^\star \widetilde{W}^\star\|_F^2 \|\widetilde{\nu}^\star\|_2^2$$

where $\widetilde{W}^\star = W^\star \sqrt{\mathrm{diag}([n_1, n_2, \cdots, n_M])}$ and $\widetilde{\nu}^\star(m) = \frac{\nu^\star(m)}{\sqrt{n_m}}$. equation 35 is derived from $\mathbb{E}\left[x_{M+1,n} x_{M+1,n}^\top\right] = I$. equation 36 is derived from Lemma C.1. equation 37 is derived from the least square estimator solution of the optimality of $\widehat{w}_{M+1}$. equation 38 is derived from $P_{X_{M+1}\widehat{B}_T}^{\perp}{}^\top P_{X_{M+1}\widehat{B}_T} = 0$. equation 39 is derived from $w_{M+1}^\star = \widetilde{W}^\star \widetilde{\nu}^\star$. Given that $Z_{M+1}$ follows i.i.d. Gaussian distribution with a zero mean and variance $\sigma^2$, it follows that $\frac{1}{\sigma^2}\|P_{X_{M+1}\widehat{B}_T} Z_{M+1}\|^2 \sim \chi^2(k)$. Applying the Chernoff bound for chi-square distribution, we have with probability at least $1 - \delta$, $\|P_{X_{M+1}\widehat{B}_T} Z_{M+1}\|^2 \leqslant \sigma^2(2k + 3\log\frac{1}{\delta})$. Following that, by combining the result obtained from Lemma C.2 along with applying the union bound, we derive that

with probability at least $1 - 2\delta - Td^{-10} - 2de^{-\frac{\delta'^2 n_{M+1}}{3S}}$,

$$\mathrm{ER}(\widehat{B}_T, \widehat{w}_{M+1}) \leqslant \frac{\sigma^2(2k + 3\log\frac{1}{\delta})}{2(1-\delta')n_{M+1}} + \frac{(1+\delta')}{2(1-\delta')^2}\mu^2 k \sigma_{\max}^{\star}{}^2$$

$$\epsilon^2\left(2N(d-k) + 3\log\frac{1}{\delta}\right)\|\widetilde{\nu}^\star\|_2^2.$$

Our objective in the remaining analysis is to determine the upper bound of $\|\widetilde{\nu}^\star\|_2^2$. Define $\epsilon^{-2} = \frac{N}{\|\nu^\star\|_2^2}$. Using a technique similar to Theorem 3.2 in Chen et al. (2022), for any $\gamma \in [0,1]$,

$$\|\widetilde{\nu}^\star\|_2^2 \leqslant \frac{2\|\nu^\star\|_2^2}{N}((1-\gamma)\|\nu^\star\|_{0,\gamma} + \gamma M).$$

By combining these results, we obtain the upper bound as

$$\mathrm{ER}(\widehat{B}_T, \widehat{w}_{M+1}) \leqslant \frac{\sigma^2(2k + 3\log\frac{1}{\delta})}{2(1-\delta')n_{M+1}} + \frac{(1+\delta')}{(1-\delta')^2}\mu^2 k \sigma_{\max}^{\star}{}^2$$

$$\epsilon^2\left(2(d-k) + \frac{3}{N}\log\frac{1}{\delta}\right)\|\nu^\star\|_2^2 s^\star.$$

For $0 < c < 1$, setting target sample size $n_{M+1} \geqslant \frac{\sigma^2(2k+3\log\frac{1}{\delta})}{2(1-c)(1-\delta')}\epsilon^{-1}$ ensures that

$$\frac{\sigma^2(2k + 3\log\frac{1}{\delta})}{2(1-\delta')n_{M+1}} \leqslant (1-c)\epsilon.$$

Define $t := \frac{(1+\delta')}{(1-\delta')^2}\mu^2 k \sigma_{\max}^{\star}{}^2 \|\nu^\star\|_2^2 s^\star$. For $C > 1$, setting source sample size $N \geqslant \frac{3C}{c}t\epsilon\log\frac{1}{\delta}$ results in

$$N \geqslant \frac{3C}{c}t\epsilon\log\frac{1}{\delta} = \frac{3\log\frac{1}{\delta}}{2(d-k)}C(\frac{2}{c}(d-k)t\epsilon)$$

$$\geqslant \frac{3\log\frac{1}{\delta}}{2(d-k)}\frac{\frac{2}{c}(d-k)t\epsilon}{1 - \frac{2}{c}(d-k)t\epsilon} = \frac{3t\log\frac{1}{\delta}}{c\epsilon^{-1} - 2(d-k)t} \tag{40}$$

where equation 40 is derived from the fact that there exists a constant $C > 1$ satisfying the inequality $\frac{x}{1-x} \leqslant Cx$ for $0 < x < 1$. Consequently, $(2(d-k) + \frac{3}{N}\log\frac{1}{\delta})t\epsilon^2 \leqslant c\epsilon$. Thus, $\mathrm{ER}(\widehat{B}_T, \widehat{w}_{M+1}) \leqslant \epsilon$ and completes the proof. □

# D ADDITIONAL EXPERIMENTS

This section presents additional numerical experiments that demonstrate the efficacy of our proposed algorithm for both known and unknown $\nu^\star$ on synthetic data. We evaluated our algorithm against three benchmark approaches: (i) the Method-of-Moments (MoM) estimator presented in Yang et al. (2020); Tripuraneni et al. (2021), (ii) the approach in Chen *et al.* Chen et al. (2022), (iii) our proposed estimator via a uniform sampling approach. We ran experiments on synthetic data, varying the number of tasks $M$, the rank $k$, and the dimension $d$. The proposed algorithm consistently outperforms all three benchmark approaches, as expected.

**Synthetic data:** The default parameter settings are defined as $n_m^i = 50, d = 300, k = 4, M = 100$. Following the same strategy as before, the entries of matrix $B^\star$ were randomly generated by orthonormalizing an i.i.d. standard Gaussian matrix, while the entries of matrix $W^\star$ for the source tasks were randomly generated according to an i.i.d. Gaussian distribution. In this experiment, by fixing all others at the default settings and only varying the rank or the number of source tasks, the matrices $B^\star$ and $W^\star$, as well as the data, are the same to guarantee that the differences are only attributable to changing the rank or the number of source task. The trends demonstrate significant dependence as the rank or the number of source task vary. The task relevance parameter $\nu^\star$ was randomly generated for the known setting, while in the unknown setting, $\nu^\star$ was assigned by allocating 20% of tasks a weight of 2, 60% of tasks set to 0, and the remaining 20% tasks to 8. The matrices

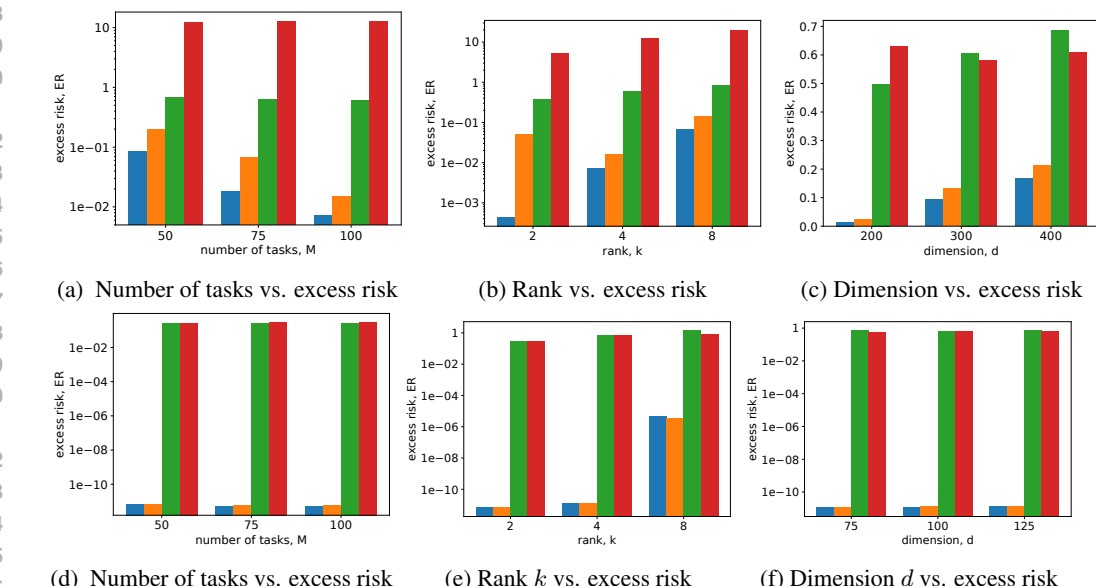

(a) Number of tasks vs. excess risk     (b) Rank vs. excess risk     (c) Dimension vs. excess risk

(d) Number of tasks vs. excess risk     (e) Rank $k$ vs. excess risk     (f) Dimension $d$ vs. excess risk

Figure 3: ■ proposed algorithm (adaptive sampling), ■ proposed algorithm (uniform sampling), ■ MoM (adaptive sampling), ■ Chen et al. (adaptive sampling). $\nu^\star$ **known setting:** Figures 3a, 3b, and 3c. $\nu^\star$ **unknown setting:** Figures 3d, 3e, and 3f. We use 200 data samples for each source task and 100 data samples for the target task. We altered the number of tasks to $M = 50, 75, 100$, altered the rank of the $\Theta^\star$ to $k = 2, 4, 8$, and altered the dimension to $d = 200, 300, 400$. It is obvious that our proposed approach with adaptive sampling outperforms all other benchmark approaches.

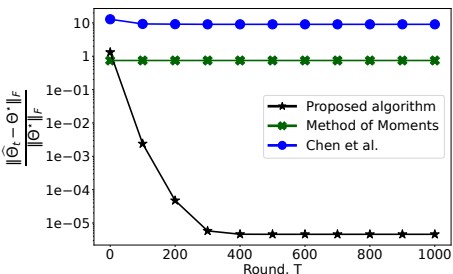

Figure 4: Estimation error vs. GD iteration for $d = 300$, $M = 100$, $k = 2$, noise variance $= 10^{-6}$.

$X_m$ were generated randomly according to an i.i.d. standard Gaussian distribution. We produced the noise following an i.i.d standard Gaussian distribution with a mean of zero and a variance of $10^{-6}$. We conducted 100 independent Monte-Carlo trials and averaged the results.

**Excess risk plots.** Figure 3 demonstrates the plots for the excess risk. Figures 3a, 3b, and 3c illustrate the excess risk for the known setting, while Figures 3d, 3e, and 3f depict the excess risk for the unknown setting. Figure 3a, 3b, and 3c demonstrate that, under the known relevance parameter setting, our algorithm consistently provides a significantly lower excess risk than the other approaches, validating the theoretical findings. In the unknown setting, as illustrated in Figures 3d, 3e, and 3f, our proposed approach significantly outperforms both the MoM estimator-based approach and the approach presented by Chen *et al.*. Our approach demonstrates significant improvement compared to the approach in the $\nu^\star$ known setting. Consequently, the numerical experiments validate our theoretical conclusions and the efficacy of the proposed approach.

**Estimation error plot ($\nu^\star$ unknown).** In Figure 4, we present the plot for estimation error vs. GD iterations. The plot clearly demonstrates that our proposed approach provides outstanding performance while achieving a very small percentage error; however, it cannot fall below the noise variance, which is set to $10^{-6}$. This validates the benefit of adaptive sampling for generalizing to a target task.

# E  ADDITIONAL RELATED WORK

**Federated learning** has emerged as a critical paradigm to address the growing demand for enhanced security and privacy in machine learning systems McMahan et al. (2017); Li et al. (2020); Bonawitz et al. (2017); Karimireddy et al. (2020). This framework ensures that raw data remains on local devices, never being shared with the central server or other tasks, thus preserving data privacy. Instead, only aggregated local model updates are exchanged, providing a privacy-preserving mechanism for collaborative learning. For detailed survey on federated learning we refer to Kairouz et al. (2021). Recently, Collins et al. (2021) proposed a federated approach for multi-task representation learning. While our work is related to that of Collins *et al.*, we distinguish ourselves by focusing on active learning and providing a convergence guarantee that explicitly accounts for initialization.

