# OpenReview forum: "PROVABLY EFFICIENT FEDERATED ACTIVE MULTI-TASK REPRESENTATION LEARNING"
_ICLR.cc/2025/Conference — Submitted to ICLR 2025_

### Official Review · Reviewer_1kUR · 2024-11-01

**Soundness:** 2
**Presentation:** 2
**Contribution:** 3
**Rating:** 5
**Confidence:** 2

**Summary:**

This work addresses the problem of multi-task representation learning for linear regression models and linear representations. The solution estimates the relevance of each source task to the target task, enabling efficient use of data from relevant source tasks. The paper also provides convergence guarantees for the algorithm, along with an analysis of its sample and time complexities.

**Strengths:**

This paper provides theoretical guarantees for the proposed representation learning approach. It has two practical improvements over existing methods (lines 170-182):

* It removes the need for knowledge of an optimal solution to the non-convex optimization cost function.

* It does not require the number of data samples to exceed the problem’s dimensionality.

**Weaknesses:**

Several aspects of the paper could be improved:

1. The scope of the paper is not clearly stated or positioned in the abstract and introduction. The work focuses specifically on linear representation functions and linear regression models; however, both the abstract and introduction discuss general low-dimensional representation and multi-task learning without mentioning this important, narrower focus.

2. The paper claims applicability to image-related learning problems. However, most of these problems, including the MNIST-C dataset used in the paper, are classification tasks typically addressed with non-linear models. This raises questions about how the proposed solution, which is designed for linear regression models, applies in such contexts. Additionally, Assumption 2.2 requires that $x_{m,n}$​ follows an i.i.d. standard Gaussian distribution, which is rarely satisfied in image-related learning tasks. How is this assumption justified for these applications?

3. The technical content of the paper is not fully self-contained, with some notations either undefined or introduced before being defined.

    a. $n_m$​ is defined for $m \in [M]$ (line 89), but the paper later uses $n_{M+1}$, which is not explained.

    b. $\Theta^*$ and $\theta^*_1,\dots,\theta^*_M$ are undefined. This makes the discussion of their SVD and their connection to $W$ and $B$ in lines 116-119 difficult to follow.

    c. The Notations paragraph (lines 129-137) appears after many of the notations have already been used. It would be clearer if this paragraph were placed at the beginning of Section 2.

    d. It is unclear how $X^{(i)}_{M+1}$ is obtained in Algorithm 1.

4. The high probability statement is presented with a probability in the form $O(1 - \delta - d^{-10} - ....)$ (lines 331) without clarifying its meaningfulness. Since the $O$-notation implies potential hidden constants, this probability may actually be small. Additionally, reducing $\delta$ does not necessarily bring this probability closer to $1$ (indicating high probability) due to the presence of other terms, particularly the exponential ones. Therefore, it is unclear if the theoretical result of this paper is an improvement over Chen et al. (2022).

**Questions:**

Apart from the weaknesses mentioned above, I have the following questions:

1. What new techniques are introduced to handle unknown relevance parameters (lines 181-182) compared to the case with known relevance parameters? The problem setup and the algorithm appear similar to those in Chen et al. (2022), aside from the cost function optimization, which applies to both known and unknown relevance parameters.

2. Is the $\epsilon$ in this paper comparable with the $\epsilon$ in Chen et al. (2022)? The discussion in lines 386-390 suggests that the two are comparable. However, in this paper, the guarantee is $ER \le \epsilon$ whereas in Chen et al. (2022) the guarantee is $ER \le \epsilon^2$.

3. In the experiments, the paper should clarify how a regression model can be applied to MNIST-C, which is a classification problem.

4. Chen et al. (2022) assumes knowledge of an optimal solution to the cost function. How is this assumption implemented in the experiments? What justifications are provided for the proposed approach, which does not rely on knowing the optimal solution but instead searches for it using GD, being able to outperform a baseline that has access to the optimal solution?

---

> ### Author Response · Authors · 2024-11-19
> **Rebuttal:**
>
> Thank you for your valuable feedback. Please see the following for our responses to your questions. Please let us know if there are any further questions.
>
> 1. **Scope:**  Our paper focuses on active multi-task representation learning with linear representation. The emphasis on linear representation is included in both the abstract and the introduction of the revision.
>
> 2.  **Nonlinear and non-i.i.d settings:**  While our algorithms and theoretical guarantees focus on linear representations and i.i.d data (Assumption 2.2), we conducted experiments using the MNIST-C dataset, which includes classification tasks typically addressed with non-linear models. This was done to assess the effectiveness of our approach in nonlinear settings. The results obtained indicate that our algorithm consistently outperformed other benchmark algorithms in these non-linear scenarios. This observation indicates that extending our approach to include non-linear models is a promising direction for future work. We have included these details in Section 6 (Simulations) and Section 7 (Conclusion and Future Work). We would also like to note that existing works, including Chen et al., Tripuraneni et al., and Collins et al., focus on independent and identically distributed (i.i.d) data, to the best of our knowledge.
>
> 3.a   $n_{M+1}$ is the total number of samples of the target task. In line 101 in the revised draft (line 89 of the previous version of the paper), we defined the number of samples for the $m$-th task as  $n_m$, for each source task $m \in [M]$. Subsequently, within the same section, in line 107, we defined that $n_{M+1}$ as the number of samples from the target task.
>
> 3.b **Definition of $\Theta^\star$:** Under Assumption 2.1 we can rewrite $y$ as $y = x^\top B^\star w_{m}^{\star} + z_m = x^\top \theta^\star_m + z_m$. Thus, $\theta^\star_m=B^\star w_m^\star$ and $\Theta^\star = [\theta_1^\star, \cdots, \theta_M^\star]$. This is included in the revision.
>
> 3.c The Notation section is now presented at the start of Section 2.
>
>
> 3.d **$n_{M+1}$ in Algorithm 1:**  $n_{M+1}$ is the total number of samples of the target task, and $n^i_{M+1}$ denotes the number of samples of the target task in the $i$-th epoch of the algorithm. We now added a step in the algorithms as ``Observe $n^i_{M+1}$ samples $X_{M+1}^{(i)}$ and $Y_{M+1}^{(i)}$ for the target task" (step 23 in Algorithm 1 and step 17 in Algorithm 2). The revised version of the paper includes this update.
>
>
> **Chen et al. vs. Our work:** While our work builds on Chen et al., 2022, our proposed approach differs from that of Chen et al., 2022. in three significant aspects.
>
> **(i)** The estimation problem for $\Theta^\star$ in Equation 2 is a non-convex optimization problem. Chen et al. assumed that either an optimal solution to this non-convex regression problem is known or that a convex (trace-norm) relaxation is considered. The theoretical guarantees provided by Chen et al. are based on the assumption that an optimal solution to Equation 2 is available. In this work, we introduce a novel alternating gradient descent algorithm to estimate $\Theta^\star$ and present our theoretical guarantees.
>
> **(ii)** The estimation of $\nu^\star$ relies on the estimate of $\Theta^\star$ (i.e., $B^\star, W^\star$) and the estimate of $w^\star_{M+1}$. In the work of Chen et al., since it is assumed that the optimal estimate of $\Theta^\star$ is known, the estimation of $\nu^\star$ is simplified to a linear regression problem. However, since $\nu^\star$ depends on the estimate of $\Theta^\star$, the effectiveness of this linear regression relies on the solution to a non-convex problem. Consequently, this leads to a temporal progression of the error, which necessitates a unique analysis and proof approach presented in our paper.
>
> **(iii)** Chen et al., 2022 considered an unlimited availability of source data; our study focuses on a data-scarce setting with a limited number of source data samples.

---

> > ### Author Response · Authors · 2024-11-19
> >
> > 4. **High probability statements:**  We have revised Theorem 5.1 and Theorem 5.2 by removing the $O(\cdot)$ notation. We present the complete form of the theorems with the exact probability values in the appendix (Appendix B and Appendix C) and a shorter version in the main paper for brevity and in the interest of space.
> > We note that all the exponential terms have a negative power, making them decay rapidly and, therefore, less significant compared to the $\delta$ dominant term.
> > We note that the theoretical results in this paper present the first complete proof since the guarantees in Chen et al. are under the assumption that the optimal solution to the non-convex optimization problem in Equation 2 is known (details are given in (i) and (ii) above).
> >
> > 5. **Unknown relevance parameters:** When the relevance parameter $\nu^\star$ is unknown, we use the least squares estimator to derive an estimate denoted as $\hat{\nu}$. In this case, the number of samples used from the $m$-th task is proportional to the relevance estimate of the $m$-th task, $\hat{\nu}(m)$. This contrasts with the scenario where $\nu^\star$ is known, allowing for the exact calculation of the required number of source data samples for each epoch.
> >
> >   The main challenge in estimating $\nu^\star$ arises because, although the estimation process for $\nu^\star$ itself is a convex problem, it utilizes the estimate of $\Theta^\star$ (i.e., $B^\star, W^\star$) and $w^\star_{M+1}$ which is obtained by solving a non-convex problem, as described in Equation 2. The key technical challenges related to the work of Chen et al. in estimating $\nu^\star$ are highlighted in the bullet points (i) and (ii)  provided in the above discussion.
> >
> >
> > 6. **$\epsilon$ parameter:** To achieve $\mathrm{ER} \leqslant \epsilon^2$, one can set the bound for $n_{M+1}$ and $N$ to $n_{M+1} \geqslant \frac{\sigma^2 (2 k + 3 \log{\frac{1}{\delta}})}{2 (1 - c) (1 - \delta^{\prime})} \epsilon^{-2}$ and $N \geqslant \frac{3C}{c} t \epsilon^2 \log{\frac{1}{\delta}}$ (lines 1423 and 1429 in Appendix C.3, revised paper).
> > Such a change, however, does not affect the result, as the $\epsilon$ in our work can be interpreted as the squared $\epsilon$ used in Chen et al.
> >  In Section 5.2, we discuss how the approach in Chen et al. necessitates an extra $\tilde{O}(M k^2 d \sigma\epsilon^{-1}\sqrt{s^\star})$ data samples, and their guarantee relies on the assumption of having knowledge of the optimal solution to the non-convex cost function. On the other hand, our proposed algorithm provides a guarantee through alternating GD and minimization estimation without depending on this strong assumption.
> >
> > 7. **Experiment using C-MNIST data:**  Inspired by the experiments in Chen et al. to evaluate the performance of our approach in real datasets (specifically, when the linear representation may not exist), we utilized the C-MNIST dataset. Although the MNIST problem is usually a classification problem with cross-entropy loss, we model it as a regression problem with $\ell_2$ loss to align with the setting studied in this paper. We modeled the regression problem by assigning each task as a binary classification task. For example, a client denoted as brightness$_0$ returns 1 if the image is digit 0 and has brightness corruption; otherwise, it returns 0. We have updated the description of C-MNIST data in Section 6.1 to clarify this point.
> >
> > 8. **Experiments:** While implementing the Chen et al. algorithm, as suggested in their works, we considered the convex relaxation of the original non-convex problem using a trace-norm relaxation and then applied projected gradient descent to derive an estimate for the rank-constrained problem. We mentioned this in Section 6.2 of the paper.  In our proposed approach, we obtain the estimate of $\Theta^\star$ using the alternating gradient descent and minimization algorithm. We note that our approach is a GD step to estimate $B^\star$ followed by a minimization step to estimate $w^\star$.

---

> > > ### Comment · Reviewer_1kUR · 2024-11-26
> > >
> > > I would like to thank the authors for their detailed response and the revised paper, which has addressed some of my concerns.
> > >
> > > Regarding the complete form of Theorem 5.1 in Appendix B, could the authors clarify why the term $- 2d \text{exp}(-\delta'^2 n_{M+1} / (3S))$ is considered negligible? Specifically, given that $\delta'$ is close to 0 and $n_{M+1}$​ is small, as the paper operates in the sparse data regime where $n_{M+1} < d$ (lines 108–114), it is not immediately apparent why this term approaches 0.

---

> ### Author Response · Authors · 2024-11-26
> **Response to Reviewer 1kUR:**
>
> Thank you for your comment. If you have additional questions please let us know.
>
> In response to your question regarding the term $- 2 d \exp(- \frac{\delta^{\prime^2} n_{M + 1}}{3 S})$ stated in the complete form of Theorem 5.1 in Appendix B, we provide the clarification here. First, we clarify that the parameter $\delta^\prime$ is introduced in Lemma C.1 to determine the bound on $X_m^\top X_m$. We note that $\delta^\prime$ does not have to be close to $0$. In Lemma C.1, we define $S$ as an upper bound for the squared norm of the data vector. Without loss of generality, we can set $S = 1$. Furthermore, the sample complexity bound for $n_{M + 1}$ has $1 - \delta^\prime$ in the denominator. As $\delta^\prime$ approaches 1, $n_{M + 1}$ increases (i.e., more number of samples), and the probability term has a larger negative value in the exponential term, leading to high probability. On the other hand, as $\delta^\prime$ decreases, the probability decreases; however, we require fewer number of target samples. Thus Theorem 5.1 captures the trade-off between probability guarantee and the sample complexity guarantee. We want to highlight that our paper is the first to provide a complete guarantee for the active low-dimensional representation learning problem.

---

> > ### Comment · Reviewer_1kUR · 2024-11-27
> >
> > I am just curious about the appropriate choice of $\delta'$. The paper states that $n_{M+1} < d$, however, as $\delta'$ approaches 1, $n_{M+1}$ will exceed $d$. Therefore, for $n_{M+1} < d$, $\delta'$ cannot be close to $1$. However, if $\delta'$ is too small, the probability guarantee may also diminish. Could the author provide clarification on selecting $\delta'$ in the context of the sparse data regime ($n_{M+1} < d$)?

---

> > > ### Author Response · Authors · 2024-11-27
> > >
> > > Thank you for your thoughtful question about the suitable selection of $\delta^\prime$.
> > >
> > > According to Theorem 5.1, it follows that $n_{M + 1}$ is at least $O\left(\frac{\sigma^2 (k + \log{\frac{1}{\delta}}) \epsilon^{-1}}{1-\delta^{\prime}} \right)$. Furthermore, according to the statement of Theorem 5.1, it is required that
> > > $\sigma^2 \leqslant \min${$\frac{c \| \theta_m^\star \|^2}{k^3 \kappa^6}, \frac{\epsilon^2 \|\theta_m^\star\|^2}{c^2 \kappa^2}$}.
> > >
> > > By substituting $\sigma^2 \leqslant \frac{\epsilon^2 \|\theta_m^\star\|^2}{c^2 \kappa^2}$ into the bound for $n_{M + 1}$, we find that $n_{M + 1}$ is at least $O\left(\frac{\epsilon \|\theta_m^\star\|^2 (k + \log{\frac{1}{\delta}})}{\kappa^2 (1 - \delta^{\prime})}\right)$. The preferred bound $\epsilon$ should be very small, allowing us to consider cases where $n_{M + 1} < d$. This will enable us to choose a suitable value for $\delta^\prime$ that guarantees the desired high probability.

---

> > > > ### Comment · Reviewer_1kUR · 2024-11-27
> > > >
> > > > Thank you for your prompt response.
> > > >
> > > > However, the result now seems quite puzzling: $n_{M+1}$ is at least $\mathcal{O}(\epsilon \dots)$. Does this imply that improving the guarantee (smaller $\epsilon$) requires reducing the sample complexity (smaller number of $n_{M+1}$)? This interpretation seems counter-intuitive.
> > > >
> > > > Additionally, it remains unclear what a suitable value for $\delta'$ is based on your explanation, such that both $n_{M+1} < d$ and a high-probability guarantee are satisfied. We also observe that the sample complexity for $n_{M+1}$ is expressed using big-O notation, while $n_{M+1} < d$ appears to be an exact requirement.

---

> ### Author Response · Authors · 2024-11-27
> **Response to Reviewer 1kUR**
>
> Thank you for your insightful comments. We apologize for any confusion that may have arisen from our earlier explanation. We clarify the three questions below and also uploaded the paper after incorporating the new changes highlighted in red.
>
>
> $\bullet$ By applying Theorem 5.3 from Lin et al. (ICML, 2024), in Lemma C.2, we can determine the bound $||\widehat{\theta}_{m, T} - \theta_m^\star|| \leqslant$
>
> $\max(\epsilon, \epsilon_{noise}) ||\theta_m^\star||$ under the condition that the noise variance $\sigma^2 \leqslant \frac{c \| \theta_m^\star \|^2}{\mu k^3 \kappa^6 n_m^1}$. Here $\epsilon_{noise} = C \kappa^2 \sqrt{NSR}$, where the noise-to-signal ratio is defined as $NSR := \frac{\sigma^2}{\min_m \|\theta_m^\star\|^2}$. In Lemma C.2 and Lemma B.3, we applied this result to simplify the result as:
>
> $||\widehat{\theta}_{m, T} - \theta_m^\star|| \leqslant \epsilon ||\theta_m^\star||$,
>
> under the assumption that $\epsilon_{noise} \leqslant \epsilon$. Applying $\epsilon_{noise} \leqslant \epsilon$ and substituting for $\epsilon_{noise}$ gives another condition $\sigma^2 \leqslant \frac{\epsilon^2 \|\theta_m^\star\|^2}{c^2 \kappa^2}$, which led to $\sigma^2 \leqslant \min$ {$\frac{c ||\theta_m^\star||^2}{\mu k^3 \kappa^6 n_m^1}, \frac{\epsilon^2 ||\theta_m^\star||^2}{c^2 \kappa^2}$}. We realized through the reviewer discussions that this leads to confusion, especially when substituting the value of $\sigma^2$ as $\frac{\epsilon^2 \|\theta_m^\star\|^2}{c^2 \kappa^2}$ in the sample complexity bound. To avoid this, we have revised the proof as follows.
> In the revision, we have the original condition $\sigma^2 \leqslant \frac{c \| \theta_m^\star \|^2}{\mu k^3 \kappa^6 n_m^1}$ from Theorem 5.3 in Lin et al. (2024) and we retain the $\epsilon_{noise}$ without further assuming $\epsilon_{noise} \leqslant \epsilon$. We have revised Lemma C.2, Lemma B.3, Theorem B.4 and Theorems 5.1 and 5.2 to reflect this. We believe this clarifies the ambiguity. If you have further questions, please let us know.
>
> $\bullet$ While we focus on data-scarce setting, it is not a requirement. Our goal was to showcase that our approach works even in settings where $n_m < d$.
> The exact expression for the bound on $n_{M + 1}$ was presented in Appendix B.2 (Proof of Theorem 5.1, line 1205). We have $n_{M+1} \geqslant \frac{\sigma^2 (2k + 3\log{\frac{1}{\delta}}) \epsilon^{-1}}{2 (1 - c) (1-\delta^{\prime})}$ which provides a basis for calculating the bound on $n_{M + 1}$. For simplicity, we presented this using big-$O$ notation in the theorem statement as $O\left(\frac{\sigma^2 (k + \log{\frac{1}{\delta}}) \epsilon^{-1}}{1-\delta^{\prime}} \right)$. In the revised version of the paper, we now provide the exact expression after removing the big-$O$ notation in the complete forms of the theorem statements (Theorems 5.1 and 5.2) presented in the appendix.
>
>
> $\bullet$ With respect to the selection of $\delta^\prime$, we analyze one of the settings from our synthetic data experiments detailed in the paper: $d = 100$, $k = 6$, $M = 80$, $n_{M + 1} = 50$, and $\sigma^2 = 10^{-6}$. Given $c = \frac{1}{2}$, $\delta = 10^{-2}$ and $\epsilon = 10^{-4}$, we select $\delta^\prime = 0.99$. The result is as follows:
> $$
> \frac{\sigma^2 (2 k + 3 \log{\frac{1}{\delta}}) \epsilon^{-1}}{2 (1 - c)(1-\delta^{\prime})} = \frac{10^{-6} \times (2 \times 6 + 3 \log100) \times 10^4}{2 \times (1 - \frac{1}{2}) \times (1 - 0.99)} = 18 < 50 = n_{M + 1} < 100 = d
> $$
> that guarantees $n_{M + 1} < d$. Furthermore, we have
> $$
> 2 d e^{-\frac{{\delta^\prime}^2 n_{M+1}}{3 S}} = 2 \times 100 \times e^{- \frac{0.99^2 \times 50}{3 \times 1}} \approx 1.6 \times 10^{-5}
> $$
> that ensure a high probability.

---

### Official Review · Reviewer_xumW · 2024-11-03

**Soundness:** 3
**Presentation:** 2
**Contribution:** 3
**Rating:** 6
**Confidence:** 3

**Summary:**

This paper is built on the premise that in multi-task learning scenarios one typically lacks a sufficient amount of data to train on. This paper addresses the data scarcity problem by developing an adaptive sampling scheme that takes the form of an alternating projected gradient descent and minimization algorithm. The algorithm iteratively estimates the relevance of a source task based on estimated relevance and then samples more heavily from tasks deemed most relevant. As is standard in the literature, the problem formulation assumes there are $M$ source tasks each with $n_m$ iid samples. In particular, the data scarcity is modeled by the fact that $n_m < d$ where $d$ is the input dimension, and $n_{M+1}\ll \{n_1, \dots, n_M\}$.  It is assumed that the underlying linear representation that is shared across all tasks is of low dimension. A convergence and sample complexity analysis shows that both scale as $\mathcal O (\log (1/\epsilon))$, moreover the number of target samples with the rank of the latent feature space $k$. The theory is reinforced through the simulations using the MNIST-C data set.

**Strengths:**

I believe the paper addresses a relevant problem, and the proposed solution performs well in theory and practice. The algorithm is somewhat intuitive and by that there are no big surprises in any steps.

- The authors have included a detailed literature review which clearly situates the problem they address
- The paper is specifically concerned with the setting where the relevance parameter $\nu^\star$ is unknown. Estimating $\nu$ introduces errors in the estimate of $\Theta$ and this requires a non-trivial analysis to ensure convergence.
- Comparing the case where $\nu^\star$ is known and contrasting the algorithmic performance is appreciated (even if this is buried - see next section).

**Weaknesses:**

My biggest concern is with the flow of the paper and the clarity of exposition. It is quite well written but there are areas that need to be improved.
- The main theoretical results of the paper are presented in Theorems 5.1 and 5.2 presented in section 5. The section 5.1 header indicates that there are two results, one for when $\nu^\star$ is unknown which corresponds to Algorithm 1 and one for  Algorithm 2 where it is known. While reading the paper, it is never even mentioned that there is an algorithm 2.
- Some of the core notation is confusing. Specifically, $\nu^\star$ is defined as a vector of dimension $M$ but also as function of $m$ on the same line.
- A significant part of the paper depends on the alternating gradient descent minimization algorithm described in the papers by Lin, Neyer, and Collins. This algorithm has been shown to badly suffer in a number of scenarios, including when source noise covariance is not described by a diagonal matrix - see for example "Sample-Efficient Linear Representation Learning from Non-IID Non-Isotropic Data", Zhang et al.

**Questions:**

- I wonder how important it is to have an accurate singular value decomposition? I'm curious to see how a randomized SVD may perform c.f. "Finding structure with randomness: Probabilistic algorithms for constructing approximate matrix decompositions", Halko et al.
- Is there anything that can be said if the "ground truth" tasks don't exactly share a common representation but instead there is a slight mismatch?

---

> ### Author Response · Authors · 2024-11-19
> **Rebuttal:**
>
> Thank you for your valuable feedback. Please see the following for our responses to your questions. Please let us know if there are any further questions.
>
> 1. **Algorithm 2:** We have updated Section 5.1 of the paper after including the details below.
>
> Algorithm 1 deals with the unknown $\nu^\star$ setting, and Theorem 5.1 presents the guarantee. Algorithm 2 (given in Appendix C) presents the $\nu^\star$ known setting, and the guarantee is given in Theorem 5.2. The primary distinction is that in Algorithm 1, the estimate of $\nu^{\star}$ relies on the estimate of $\Theta^\star$ (i.e., $B^\star, W^\star$) and the estimate of $w^\star_{M+1}$, which introduces a temporal effect in error propagation. In contrast, Algorithm 2 operates under the condition that $\nu^{\star}$ is known. In both cases, we demonstrate that, given the suitable sample complexity conditions for both the source and target tasks, the excess risk is bounded by $\epsilon$ with high probability. In the interest of space, we present Algorithm 2 in the appendix.
>
> 2. **Notation:** We believe the reviewer is pointing to line 125 (submitted version), which is line 138 in the revised version. Yes indeed, $\nu^\star$ is a vector of dimension $M$ where $\nu^\star(m)$ indicates the m-th element of $\nu^\star$. We added a detail in the revision to clarify this.
>
> 3. **Non-iid:** We thank the reviewer for pointing us to this recent work. The guarantee in the paper, however, is not complete, as mentioned in Remark 3.2 of Zhang et al. The main result in the paper (Theorem 3.1) relies on the assumption that the representation of the initialization step is sufficiently close to the ground truth.  However, there is no guarantee of initialization provided. To this end, we are unable to use any results directly from Zhang et al. We plan to investigate the non-i.i.d setting as part of our future work.
>
> 4. **Singular value decomposition:** We believe the reviewer is pointing to the use of $k$-SVD in the initialization step to extract the singular vectors.
>  In our approach, the time complexity of the $k$-SVD step $d M k$ times the number of iterations required. We notice that to obtain an initial estimate of the span of $B^{\star}$ that is $\delta_0$-accurate, where $\delta_0 = \frac{c}{\kappa^2\sqrt{k}}$, it is sufficient to use an order $\log(\kappa k)$ number of iterations. Thus, since $n_m^1 \geqslant k$, the total complexity of the initialization phase is $O(d (\sum_{m=1}^M n_m^1 + M k) \log (\kappa k)) = O(\sum_{m=1}^M n_m^1 d \log \kappa k)$.
> Thus, the complexity term of $k$-SVD is subsumed by the computational cost of the other steps. To this end, there is no major computational advantage of using a randomized approach for SVD rather than the $k$-SVD in our algorithm.
> This detail is provided in Section 5.2 of the paper.
>
> 5. **Common representation:** Theoretical guarantees in the paper require the tasks to share a low-dimensional common representation. Using the corrupted MNIST data (C-MNIST), we demonstrate that our proposed approach performs well and surpasses state-of-the-art methods, even when a low-dimensional linear model and a common representation do not exist. We noted this in the revised paper (Sections 6 and 7). A detailed analysis of nonlinear models will be a topic for future work.

---

> > ### Comment · Reviewer_xumW · 2024-11-26
> >
> > I thank the authors for their responses and clarifications. I will maintain my original score.

---

### Official Review · Reviewer_yeea · 2024-11-05

**Soundness:** 3
**Presentation:** 2
**Contribution:** 2
**Rating:** 6
**Confidence:** 3

**Summary:**

This paper studies efficient active learning in a multitask setting, where the goal is to learn a shared representation and a target linear predictor. Data from source tasks are actively sampled by discovering the relevance of each source task to the target task. The proposed method adopts alternating gradient descent to obtain the shared linear representation.  Theoretical analysis on the convergence and generalization guarantees are provided. The proposed method is tested with numerical experiments on synthetic data and MNIST.

**Strengths:**

This paper introduce an active multi-task learning theoretical  framework. The proposed method is based on intuitive idea, i.e., essentially eq (4) where it adapts the relevance estimate of each source task to the target task. Although the idea is intuitive, the technical details, both algorithmically and theoretically, are non-trivial. Numerical experiments verifies the effectiveness of the proposed method on simple cases with linear models.

**Weaknesses:**

* There seems to be an important assumption that is not clearly elaborated. The introduction of the condition number $\\kappa$ at line 119 essentially requires $\\Sigma$ is full-rank. In other words, it is assumed that the the target task $\omega^\star\_{M+1}$ is always a linear combination of  the set of  $\\{\omega\_m\\}\_{m\in M}$ corresponding the $M$ tasks. This is necessary to require $M\\geq k$ where $k$ is the data manifold dimension, i.e., there must be enough number of source tasks. This seems to be a rather strong assumption, and it seems that it can be relaxed, e.g., suppose there is only a few source tasks but they are all close to the target task, so the learned representation should transfer to the target task even if $\omega^\star\_{M+1}$ is not a linear combination of  the set of  $\\{\omega\_m\\}\_{m\in M}$. Can the authors elaborate more on this assumption and how it may be alleviated?
* The paper has the term "Federated" in the title, suggesting a federated setting, but it seems that it is not really a federated learning paper. e.g., the word "federated" is only mentioned twice in the main paper. I understand that the proposed method can be directly applied in a federated setting, but it seems a misleading to emphasize this in the title. Or more discussion about the application to federated learning, including relate work, should be included.
* It would be nice if there are discussions on how the proposed method may be applied to non-linear settings.
* There are many typos and the presentation could be improved. For example, the matrix $W$ is used at line 116 before its actual definition at line 137. The parameter $\beta$ at line 222 seems to be undefined? Is the $\epsilon_3$ at line 331 a typo? Theorem 1 and 2 are quite hard to read and some more discussion to introduce each part before the theorems should be better, e.g., think of a sketched version of the theorem.

**Questions:**

* see previous section for questions

---

> ### Author Response · Authors · 2024-11-19
> **Rebuttal:**
>
> Thank you for your valuable feedback. Please see the following for our responses to your questions. Please let us know if there are any further questions.
>
> 1. **Rank assumption:** To clarify the above question, we now included rank$(\Theta^\star)=k$, where $k \ll \min \{d, T\}$ in the paper (line 129 in the revised paper).
>
> As discussed in the paragraph below Assumption 2.1, we consider feature matrix $\Theta^\star$ is rank-$k$. The definition of the condition number $\kappa$ hence follows.
> To this end, the feature of the target task $w^\star_{M+1}$ is a linear combination  of the source tasks $w^\star_{m}$, where $m=1, \ldots, M$.
> The low-dimensional assumption captures the relatedness between the tasks and is used in many works on representation learning, including  Chen et al., 2022, Du et al., 2020, Tripuraneni et al., 2021, Yang et al., 2020, Hu et al., 2021, Cella et al., 2023, and Kumar et al., 2022.
>
> If the number of source tasks is insufficient, it can negatively impact the quality and generalizability of the learned representations. The representation learning literature typically focuses on scenarios with source tasks that can provide a solid foundation for effective transfer learning and representation learning. In our work, we present an approach with theoretical guarantees on transferability to the target task. Further, we relax the assumption on the requirement on the number of source task data samples in the literature.
>
> 2. **Federated:** The proposed approach and algorithms are indeed a federated approach since in collaborative learning, the agents/tasks do not share the raw data; rather, only their estimates (the estimate of $B^\star$) are shared with the central server. In the revised paper, we mention this in the introduction. Also, we have now included a related work discussion on federated learning in the appendix. We are also open to modifying the title to ``Provably Efficient Multi-Task Representation Learning" if allowed.
>
> 3. **Nonlinear settings:** While our algorithms and theoretical guarantees focus on linear representations, we conducted experiments using the MNIST-C dataset, which includes classification tasks typically addressed with non-linear models. This was done to assess the effectiveness of our approach in nonlinear settings. The results obtained indicate that our algorithm consistently outperformed other benchmark algorithms in these non-linear scenarios. This observation indicates that extending our approach to include non-linear models is a promising direction for future work. We plan to investigate this in our future studies. We have included these details in Section 6 (Simulations) and Section 7 (Conclusion and Future Work).
>
> 4. **Typo and presentation:**  We have made revisions to the paper to improve clarity. We moved the notation paragraph to the beginning of Section~2 to address the concern. Additionally, we now defined the parameter $\beta$ in the algorithm, which was missing. Regarding $\epsilon_3$ at line 331 (submitted version), it was correct as stated. This parameter was introduced to derive the probability from Bernstein's inequality, satisfying $\epsilon_3 \leqslant 1$ in Lemma B.2. However, we have updated the proof, specifically the probability to avoid new notations (please see Appendix B). Finally, we have provided a short version of the theorem in the main paper to enhance clarity, and we present the complete form in the appendix. The updated version has been uploaded for your review.

---

> > ### Comment · Reviewer_yeea · 2024-11-26
> >
> > I appreciate the response and clarification. I would keep my overall rating unchanged.

---

### Author Response · Authors · 2024-11-25
**Follow-Up on Rebuttal:**

Dear Reviewers,

Thank you once again for your valuable feedback. We have carefully addressed all the comments and concerns raised in the reviews and have uploaded the revised manuscript for your consideration. We would like to kindly request that you take another look at our responses to ensure that all your concerns have been adequately addressed. If there are any additional questions, please let us know. Thank you.

---

### Meta-Review · Area_Chair_G6vB · 2024-12-19

**Metareview:**

Paper studies active learning for multi-task learning through shared linear representation across tasks. Study of active learning is a novel. Although, it is true that some previous works have mentioned this kind of work is federated, but neither theory nor experiments discuss this connection in this paper, hence the title is misleading. Some of the assumptions seems too strict. Paper lacks clarity in exposition.

**Additional Comments On Reviewer Discussion:**

Authors addressed some of the reviewer comments, but reviewers were not fully satisfied. Authors are encouraged refine the manuscript to address these concerns.

---

### Decision · Program_Chairs · 2025-01-22

Reject